# Fjord circulation permits persistent subsurface water mass in a long, deep mid-latitude inlet

Laura Bianucci[1], Jennifer. M. Jackson[1,2], Susan E. Allen[2], Maxim V. Krassovski[1], Ian J.W. Giesbrecht[3], and Wendy C. Callendar[1]

[1]Institute of Ocean Sciences, Fisheries and Oceans Canada, Sidney, BC, Canada
[2]The University of British Columbia, Vancouver, BC, Canada
[3]Hakai Institute, Vancouver, BC, Canada

*Correspondence to*: Laura Bianucci (laura.bianucci@dfo-mpo.gc.ca)

**Abstract.**

Fjords are deep nearshore zones that connect watersheds and oceans, typically behaving as an estuary. In some fjords strong katabatic winds in winter (also known as Arctic outflow wind events) can lead to cooling and reoxygenation of subsurface waters, with effects lasting until the following autumn, as observed in 2019 in Bute Inlet, British Columbia, Canada. We used high-resolution, three-dimensional ocean model summer simulations to investigate the mechanisms allowing for the persistence of these cool, oxygen-rich subsurface conditions in Bute Inlet. The slow residual circulation underneath the brackish outflow (and consequent slow advection) in this long, deep fjord is a main reason why the cold subsurface water mass stays in place until conditions change in autumn (i.e., start of stronger wind mixing and reduced freshwater forcing). Another mechanism is a positive feedback provided by the presence of this subsurface water mass, since it further reduces the already weak residual circulation. These findings are applicable to any similar long, deep fjord that experiences katabatic winds in winter, and they could have implications not only for the preservation of water masses but other possible subsurface features (e.g., pollutant spills, planktonic larvae). Furthermore, the identification of mechanisms that permit persistent cold and oxygenated conditions are key to understanding potential areas of ecological refugia in a warming and deoxygenating ocean.

## 1 Introduction

High- and mid-latitude coastlines are beset with innumerable fjords, remnants of glacial periods. The importance of these coastal geomorphological features extend to many realms, since they provide habitats for multiple species (Arimitsu et al., 2012; Keen et al., 2017; Mathews and Pendleton, 2006; Frid et al., 2021), receive inputs from both the watersheds and the neighbouring ocean (Bianchi et al., 2020; St. Pierre et al., 2022), and offer protected waters for transportation, aquaculture, fisheries, and other human activities (Iriarte et al., 2010; Bergh et al., 2023). They are also important sites for traditional cultures (e.g., Ball, 2021; Brattland, 2010). Global organic carbon burial rates in fjords are disproportionately large for their small area size (Smith et al., 2015) and their high sedimentation rates allow for high-resolution records of past climates (Bianchi et al., 2020). Fjords properties have been changing in the last several decades showing, for instance, evidence of warming and

deoxygenation in several parts of the world (Aksnes et al., 2019; Jackson et al., 2021; Linford et al., 2023). Their hydrological forcing will change under warmer climates, as runoff timing shifts to higher winter flows and earlier snowmelt freshet (e.g., Bidlack et al., 2021) and glacial runoff reaches a peak in the future, to later decrease (e.g., Rounce et al., 2023). While further changes are to be expected, it is still uncertain how climate change will affect current fjord ecosystems.

One of the fjords showing long-term trends is Bute Inlet (Jackson et al., 2021), a mainland fjord in British Columbia (BC, Canada; Fig. 1b) that lies within the traditional territory of several First Nations: Homalco, We Wai Kai, We Wai Kum, Kwiakah, and Tla'amin Nations. It is about 80 km long and 3 km wide, with a maximum depth of 730 m and a single 355 m sill at its mouth (Pickard, 1961). Approximately 94% of the freshwater entering Bute Inlet is supplied by two major rivers near the head (Homathko and Southgate, Fig. 1a) while the rest of the freshwater is provided by small streams (Farrow et al., 1983).

A recent study determined that about three times more terrestrial organic carbon per unit of surface area is buried in Bute Inlet sediments each year compared with other studied fjords (Hage et al., 2022), suggesting that this inlet is an important conduit for land-sea exchange as well as a region with high primary productivity. The high rates of terrestrial carbon burial are partly due to the frequent turbidity currents in Bute Inlet (Heijnen et al., 2020; Hage et al., 2022), which are associated with river discharge and tides (Bailey et al., 2023). Effects of climate change are already clear in Bute Inlet; for instance, its deep waters

(i.e., below sill depth) warmed by 1.3°C and lost about 0.6 mL L$^{-1}$ of oxygen from 1951 to 2020 (Jackson et al., 2021). Furthermore, a landslide in November 2020 caused by rapid deglaciation in a Southgate River tributary created a land-based tsunami and subsequent outburst flood; the generated sediment plume was observed in Bute Inlet more than 60 km from its source (Geertsema et al., 2022).

Despite the observed warming and deoxygenation at depth, recent observations in Bute Inlet showed that subsurface (i.e.,

above the sill depth) cold and oxygen-rich waters originated during an Arctic outflow wind event in February 2019 and persisted until the following fall (Jackson et al., 2023). These authors determined that the strong, cold February katabatic winds mixed the top ~100 m of the water column in Bute Inlet, leading to cooling and oxygenation down to that depth; they also observed temperature minima and oxygen maxima in monthly vertical profiles from March until October 2019. Such subsurface temperature minima had been previously observed in BC fjords and suggested to be associated to Arctic outflow

events (MacNeill, 1974; Pickard, 1961). The year-long permanence of such subsurface characteristics could alleviate deteriorating conditions and/or provide refugia to multiple species as climate changes (Ducklow et al., 2022; Jackson et al., 2023). How these winter-generated subsurface conditions can remain unchanged for almost a whole year has not yet been studied.

Here, we used a high-resolution three-dimensional numerical ocean model to investigate the mechanisms responsible for the

lingering of the 2019 cold/oxygenated subsurface feature in Bute Inlet. We analyzed summer simulations with and without the subsurface cold feature, describing the along-fjord circulation under both conditions. Tracking of Lagrangian particles further allowed us to explore the retentiveness of the fjord. Finally, we discussed how the slow circulation typical of such a deep and long fjord partly explains why a subsurface feature can persist. The latter mechanism was aided by a positive feedback, since the presence of the subsurface cold water mass led to even slower residual circulation underneath the surface estuarine outflow.

Thus, a subsurface feature is able to stay mostly in place until conditions change in autumn (i.e., stronger winds and reduced freshwater forcing after peaking during the summer freshet).

## 2 Methods

### 2.1 Model grid, parameterizations and parameter values, and particle tracking

The model used for this work evolved from the Finite Volume Community Ocean Model (FVCOM; Chen et al., 2003, 2006)
application for the Discovery Islands developed by Foreman et al. (2012), herein referred to as F12. For the present work, the FVCOM code was upgraded from version 2.7 to 4.1 (Chen et al., 2013). The grid was mostly the same as in F12 (horizontal resolution from ~18 m in the narrowest channels to ~1.2 km at the Strait of Georgia boundary; resolution calculated as the square root of twice the element area), except for a refinement in Bute and Toba Inlets to improve the representation of the flow in those deep, narrow, and steep-sided fjords. The model bathymetry in these inlets was regenerated using a 10-meter
resolution digital elevation model (Government of Canada, 2021), and the model grid excluded the steepest sections of the near-shore while also increasing the resolution, particularly in Bute Inlet (mean resolution of 182 m instead of 351 m as in F12). The resulting grid exhibited a total of 39,532 nodes and 72,518 elements (Fig. Figure 11a). The model bathymetry was smoothed with a volume preserving technique that limits the ratio $\Delta h/h<0.1$ within each triangle in Bute and Toba Inlets; everywhere else, the threshold was 0.3. The number of vertical terrain-following layers increased from 20 to 40, keeping higher
resolution near the surface (surface layer represented 0.1% of the total water column; the coarser layer at the bottom represented 7.5% of the total depth).

Only a few parameterizations and parameter values changed in the new version of the FVCOM Discovery Islands model with respect to F12. The new setup used the k-ε vertical turbulence closure scheme (Rodi, 1987) and a background vertical diffusion and viscosity of $10^{-5}$ $m^2$ $s^{-1}$. The remaining parameterizations and parameters stayed the same and most are listed here for
convenience. For instance, the Smagorinsky eddy parameterization was used for horizontal diffusivity with a coefficient C=0.02 and the horizontal viscosity was 10 times the diffusivity. Furthermore, the bottom roughness equation remained based on the General Ocean Turbulence Model (Burchard and Bolding, 2001) with a length scale of $10^{-3}$ m and a minimal value of $2.5\ 10^{-3}$ for the model bottom drag coefficient. The external and internal time steps were kept at 0.075 and 0.75 seconds (i.e., ISPLIT parameter set to 10). At the two open boundaries (one in the Johnstone Strait and the other in the Strait of Georgia,
Fig. 1a), implicit radiation conditions were used for temperature and salinity (Blumberg and Kantha, 1985) and clamped conditions for surface elevation (Beardsley and Haidvogel, 1981); moreover, a sponge layer of 7.5 km wide was set with a damping coefficient of $1.5\ 10^{-4}$.

The Lagrangian particle tracking module developed by Chen et al. (2006, 2013) was adapted at Fisheries and Oceans Canada to run offline, using hourly velocity outputs from FVCOM. This computationally efficient particle tracking model is called
PTrack and simulates the particle trajectory until any of three termination conditions occur: advection outside the model domain, encountering land ("grounding"), or exceeding a user-imposed tracking limit (the latter was not implemented in this

study). This Lagrangian model has been used for many applications related to tracking of pathogens, viruses, salmon post-smolts, etc. (e.g., Foreman et al., 2015; Quinn et al., 2022; DFO, 2022). Here, we followed the application of PTrack described in a study of the hydrodynamic connectivity between marine finfish aquaculture facilities in BC (DFO, 2022), using a time step of 10 seconds and outputting particle locations every hour.

## 2.2 Model forcing and initialization

The external forcing and initialization changed substantially from the previous versions of this model. First of all, the time period of simulation changed from April 2010 (F12) and April - October 2010 (Foreman et al., 2015) to May - June 2019, given the focus on understanding the stability of the subsurface temperature minimum found that year in Bute Inlet (Jackson et al., 2023).

Furthermore, the present setup benefited from new operational models to force the boundaries of the FVCOM domain. At the surface, hourly forcing of winds, pressure, and heat and water fluxes were provided by the High Resolution Deterministic Prediction System (HRDPS). The first release of HRDPS offered a resolution of 2.5 km (Milbrandt et al., 2016), which was suboptimal for a region with narrow inlets (~1 km wide) such as the Discovery Islands area; therefore, for this study we used the experimental HRDPS version at 1 km horizontal resolution (MSC Open Data, 2022). The SalishSeaCast model (Soontiens et al., 2016; Soontiens and Allen, 2017), a three-dimensional physical-biological-chemical ocean model for the Strait of Georgia and Salish Sea, provided hourly temperature, salinity and surface elevations for both the Johnstone Strait and Strait of Georgia open boundaries. Moreover, temperature and salinity fields from SalishSeaCast were used to initialize the FVCOM simulation, except in Bute Inlet, where observed profiles from 23 May 2019 allowed for the best possible initialization of the subsurface temperature minimum.

The 11 rivers included in the model (Fig. 1a) were implemented analogously to F12, except that the temperature and salinity at the discharge nodes were calculated using a mass conservation approach rather than specifying the actual value (i.e. RIVER_TS_SETTING parameter was set to 'calculated' instead of 'specified'). As in the previous study, only four of the 11 rivers were gauged by the Water Survey of Canada during 2019 (Homathko, Campbell, Salmon and Oyster Rivers) and for the seven ungauged rivers, a watershed area-ratio approach was used to estimate their discharge. In F12, all ungauged rivers were estimated as the Homathko River discharge multiplied by their watershed area ratio (i.e., area of ungauged river divided by area of Homathko River). However, the new version of the model took advantage of a recent watershed characterization and classification (Giesbrecht et al., 2022) to select a representative donor gauge for each ungauged watershed. Specifically, each ungauged watershed was assigned a donor gauge from a nearby watershed of the same type and with similar climate and hydrology. Hence, while the Southgate and Toba Rivers still used the Homathko River (all three having a glacierized mountain watershed type), the Stafford, Apple, Phillips, and Brem Rivers based their discharges on the Wakeman River (snow mountain watershed type). The Powell River was estimated from the Campbell River, given that both are snow mountain type watersheds where discharge is controlled by dams (i.e., the underlying assumption being that both rivers were managed in the same way).

### 2.3 Model simulations

The model was run for ~1 month, from 24 May to 27 June 2019. The choice of this period was determined by several factors. First of all, observations from summer 2019 in Bute Inlet showed evidence of the subsurface temperature minimum (and oxygen maximum) generated the previous winter during an Arctic outflow wind event (Jackson et al., 2023). Observations in Bute Inlet from 23 May (eight profiles) provided the temperature and salinity initial conditions in the fjord. Moreover, HRDPS-1km outputs were available starting in 24 May 2019 (i.e., limiting any potential earlier start date). The total length of the

simulation allowed for 5 days of spinup and 29 days for analysis; the latter is an appropriate averaging period to remove tides and calculate residual flows (Foreman et al., 1992). Longer simulations were not pursued partly because of the diffusive nature of FVCOM, which makes it challenging to reproduce a deep and narrow fjord without data assimilation. Nevertheless, the chosen month properly represented the summer conditions in the inlet, since the freshwater forcing was high from late May to late September, the wind conditions were stable (blowing mostly from the south until September), and the values at the open

boundaries were similar throughout the summer (see Appendix B).

To represent idealized summer conditions in the absence of strong deep winter mixing the previous winter (e.g., by an Arctic outflow wind event), a sensitivity experiment was performed by removing the temperature minimum feature in Bute Inlet from the initial conditions. This experiment represents an extreme scenario, given that strong Arctic outflow wind events are common in winter in this region (more than 2 events per year on average; Jackson et al., 2023), such that some degree of

subsurface cooling is usually present (e.g., MacNeill, 1974; Pickard, 1961). All other initial conditions and forcings (e.g., atmospheric and open boundaries) remained unchanged, given that the winter deep-mixing event would only affect summer conditions in the fjord (e.g., summer open boundary conditions in the Strait of Georgia and Johnstone Strait would not be affected by the outflow winter event in Bute Inlet). It is unclear how the winter event might have affected the summer river discharge; we kept this forcing unchanged to focus on the role of the initial conditions, acknowledging that this assumption is

a source of uncertainty. In this sensitivity simulation, initial temperature and salinity profiles in Bute Inlet were kept constant below the main pycnocline (Fig. A1); the constant values corresponded to the coolest and saltiest observations in the deepest third of the water column.

The same particle tracking experiment was run with velocity outputs from each simulation (referred to as "baseline" and "sensitivity" depending on whether they were initialized with the observed profiles in Bute Inlet or not, respectively). Virtual

particles were released at the start of both simulations, using the same initial locations in both cases. Namely, particles were released within the location of the observed cold subsurface feature, i.e. at every grid node in Bute Inlet where temperature was ≤ 8 °C in the baseline initial conditions. Particles were tracked during the whole length of the simulations as they were advected by the 3D flow fields; if particles reached the coastline or seafloor at any given time, they became "grounded" and were removed from the particle count. Time series of the non-grounded particles remaining in Bute Inlet were calculated. We

had the ability to let particles trapped in the bottom bounce back into the water column if the bottom vertical velocity was upwards (i.e., include particle resuspension); however, results did not change significantly and are not shown.

## 2.4 Available observations and metrics for model evaluation

Observed vertical profiles were available for model evaluation from bottle and Conductivity-Temperature-Depth (CTD) measurements. In total, 114 profiles were available for the period 24 May - 27 June (see locations in Fig. 1b), representing 20,348 matches between modelled and observed temperatures (20,231 matches for salinity). Sea surface elevations were available at the Campbell River tidal gauge (triangle in Fig. 1b). Unfortunately, no observed velocity profiles were available in Bute Inlet to evaluate the modelled currents.

To evaluate the performance of the model, potential temperature ($\theta$) and salinity fields were compared against observed values. For this purpose, model outputs were selected from the grid node closest to each observation and linearly interpolated in the vertical dimension and time to create model-observation pairs for each available in situ sample. With all the available pairs, we calculated several metrics frequently used to quantify model-observations misfit (both for the full model domain and for the top 100 m only). The model bias determines the mean deviation between modelled and observed values, while the root mean square error (RMSE) measures the deviation in a least-squares sense. These two metrics retain the units of the analysed variable. Two useful nondimensional metrics are the model efficiency (ME) or skill and the Willmott skill score (Willmott, 1981). The former relates the deviation between model and observations to the variability in the observations, while the latter is an indication of the model error divided by the range of the observations. Both skill metrics can range from zero to one, with one indicating perfect agreement between observations and model. The latter is also the case for the square of the Pearson's correlation coefficient ($R^2$), which quantifies the correlation between modelled and observed data. A succinct but thorough description of these metrics can be found in Lehmann et al. (2009) and Liu et al. (2009).

## 3 Model evaluation

The model performance was evaluated through both quantitative and qualitative approaches. The quantitative metrics showed a good agreement between model and observations, both for the whole model domain and for the upper 100 m (Table 1). For the whole water column, biases were less than one tenth for both temperature and salinity (0.08 ºC and 0.05 g kg⁻¹, respectively), while RMSEs were below 0.5 ºC and 0.8 g kg⁻¹. Bias and RMSE values were somewhat larger if only the top 100 m were considered (Table 1), given the larger range of conditions in the upper layers; for instance, the model has trouble representing the fresh/brackish waters in Bute Inlet (Fig. 2b). All non-dimensional metrics were at or above 0.8, with particularly high Willmott skill scores above 0.92 for both temperature and salinity. Metrics for sea surface height at Campbell River were also good, with bias and RMSE values of -16 and 21 cm, respectively, and non-dimensional metrics above 0.95 (Table 1; Fig. A2).

Two-dimensional histograms (Fig. 2a and b) showed that most of the model-observations pairs fell right on top of the 1:1 slope for temperature and salinity. The least square linear fit (dashed black line) for temperature had a slope of 0.9, quite close to the desired slope of one (Fig. 2a). For salinity, the spread at low observed values led to a less successful fit, with a slope of 0.6 (Fig. 2b). The model could not achieve the low salinities observed at surface in many inlets and channels with freshwater

inputs, probably due to numerical overmixing. Nevertheless, low salinity plumes were represented, albeit not reaching values as low as observed (see salinity profiles in Fig. 3). Furthermore, the model captured the statistical characteristics of the observations, shown by the overlap of model and observed histograms (Fig. 2c and d).

The observed temperature profiles in Bute Inlet in June 12 and 26 showed a temperature minimum around 80 m depth, which was also present in the model results, albeit somewhat shallower (~45 m) and not as sharply defined (Fig. 3a, b). The latter is likely due in part to numerical mixing, since horizontal diffusion in FVCOM occurs parallel to the sigma layers (Chen et al., 2006), a simplification that can lead to an overly diffusive model in regions with steep topography and significant slopes in the terrain-following layers (Foreman et al., 2023). At the location of the observed temperature minimum, salinity and density showed distinct vertical gradients (Fig. 3c-f); these also were diffused in the model. However, the observed temperature and salinity features compensated each other in density, such that overall, the model was better able to represent the density structure at this depth (Fig. 3e, f). As mentioned before, the model overestimated surface salinity (by several g kg$^{-1}$; Fig. 3c, d), leading also to an overestimation of surface density (Fig. 3e, f). Nevertheless, both the main halocline and pycnocline were correctly represented by the model, with a sharp vertical gradient in the top 20 m of the water column. Bottom values were homogeneous and matched the observations below ~300 m. The strong resemblance of the main halocline and pycnocline (both in the observations and the model) highlights the dominant role of salinity in the stratification of the region (i.e., a beta ocean; Carmack, 2007); clearly, a subsurface temperature minimum is only possible if salinity drives density.

## 4 Along-inlet temperature and circulation

### 4.1 Baseline simulation with observed initial conditions

Temperature and along-inlet velocity were averaged over the last 29 days of the simulation. This averaging effectively filtered out the tides (to focus on residual flows) while also removed the first five days of simulation to allow for spin-up. A transect plot of mean along-inlet velocities through Bute Inlet showed a multi-layered structure of the velocity field in most of the fjord (Fig. 4a). The surface layer flowed outwards of the fjord, with a return flow underneath down to approximately the depth of the outside sill (~300 m), following a typical estuarine circulation. However, the return flow had a clear vertical structure, with velocities close to zero at the depth of the minimum averaged temperature (Fig. 4a and b; a dotted horizontal line at 50 m highlights the co-location of the near-zero averaged velocities and the mean temperature minimum). Below the depth of the outside sill, the mean, slow flow was towards the mouth of the inlet, with a narrow and weak inflow layer near the seafloor. At around 70 km from the head of the inlet, mean velocities showed the effect of tidal mixing over the outer sill (Fig. 4a) given the strong tidal currents in the Discovery Islands region (Foreman et al., 2012, 2015).

To further explore the vertical structure of potential temperature and along-inlet velocity, profiles were plotted every 10 km in the middle of the inlet (from 20 to 50 km away from the head of the inlet; Fig. 5a to d). Four or five layers are evident in the mean along-inlet velocity profiles when considering the return flow between ~5 and ~300 m depth to be divided in two where the velocities approach zero, i.e. above and below the depth of the temperature minimum feature. The temperature minima in

each profile is found between 44 and 46 m deep, with mean velocities there ranging between -0.01 and -0.7 cm s$^{-1}$ (negative values imply flow towards the head of the fjord).

## 4.2 Sensitivity simulation without temperature minimum feature

A sensitivity simulation with homogeneous conditions under the main pycnocline (as described in section 2.3) showed a single
230    return flow from underneath the surface outflow to the depth of the outside sill (Fig. 4d and Fig. 5e to h), instead of the two return layers found at those depths on the baseline simulation. The surface, outward-flowing layer was almost identical in both simulations (Fig. 4 and 5) and the bottom outward and weak inward near-seafloor flows were quite similar (e.g., comparable magnitude, vertical distribution/shape, and depth range). Ignoring the weak bottom mean inflow between 40 and 60 km, the vertical mean flow structure in this simulation could be described as three-layered (Fig. 5e to h).
235    The cold subsurface feature can be identified as a new water mass when comparing temperature-salinity diagrams for both the baseline and sensitivity simulations (Fig. 6a; only model values at the time and location of the observations were plotted to simplify the figure). The strong winter mixing event not only led to subsurface cooling and reoxygenation, but also affected salinity (see profiles in Fig. 3c,d) and led to density and stratification changes when comparing both simulations (Fig. 6c,d). In particular, stratification decreased below the outward-flowing layer between ~5 and 50 m in the baseline (negative values
240    in Fig. 6d) and density increased in that same region (positive values in Fig. 6c). These changes were not uniform in the horizontal, but stronger near the head of the inlet (seen in Fig. 6c and d and highlighted by the Δρ profiles at 20 and 40 km in Fig. 6b); given the overall horizontal density gradient near the surface (lower density near the head due to the freshwater inputs), the subsurface water mass led to a reduced horizontal density difference between the head and the mouth of Bute Inlet. The latter in turn led to an even slower mean along-inlet velocities underneath the estuarine surface outflow in the baseline
245    simulation.

## 4.3 High retention of subsurface particles

The slow velocities below of the surface outflow seen in both the baseline and sensitivity simulations (Figs. 4a,d and 5) also led to high retention in Bute Inlet in both particle tracking experiments (Fig. 7). The time series of the percent of particles moving inside the fjord (i.e., not grounded) showed that more than 96% of the moving particles stayed within the inlet (north
250    of 50.45 °N) by the end of the simulations. Particularly, the retention was higher in the baseline simulation (thicker line in Fig. 7), with more than 97% of the particles remaining in the inlet. Furthermore, the median depth of the particles in the baseline simulation stayed within 45 and 54 m (comparable to the initial median depth of 54 m), contrasted with the deeper median depth of the sensitivity simulation (> 100 m after 5 days, thinner line in Fig. 7). The weaker stratification below the main pycnocline led to a larger range of particle dispersion and more grounding in the latter simulation.

## 5 Discussion

The modelled circulation in Bute Inlet, as observed in many other fjords, is estuarine and multi-layered (e.g., Baker & Pond, 1995; Stacey & Gratton, 2001; Valle-Levinson et al., 2007, 2014; Wan et al., 2017). The overall summer residual circulation in the deep Bute Inlet is relatively slow, with time-averaged flows below the surface under 5 cm s$^{-1}$ (Figs. 4 and 5); these values are comparable with some fjords, e.g. Reloncavi fjord in Chile (Valle-Levinson et al., 2007, 2014) and Knight Inlet in BC (Baker and Pond, 1995), but lower than many others that easily reach or exceed 10 cm s$^{-1}$, e.g. Saguenay fjord (Stacey and Gratton, 2001), Douglas Channel (Wan et al., 2017), Sermilik fjord (Jackson and Straneo, 2016), and Aysen fjord (Valle-Levinson et al., 2014). The slow circulation in Bute Inlet is partly due to the length of the inlet, since longer distances decrease the pressure gradient between the fresh head of the inlet and the saltier mouth. Furthermore, the significant depth of the sill (~300 m) also contributes to a slow return flow, given the large associated cross-inlet area available to compensate the surface volume outflow. The slow residual velocities below the surface lead to low advection, long transit times, and an overall high retention of particles seen in our (summer) model simulations. Therefore, we identify the geometry of a long, deep inlet with freshwater forcing at its head as a main mechanism leading to the lingering of a subsurface feature. The latter feature could be a distinct water mass, as in the current study case of the cold, oxygenated waters in Bute Inlet in 2019, or potentially pollutants, microplastics, larvae, etc. somehow released under the surface outflowing layer.

A second mechanism is a positive feedback related with the existence of the subsurface water mass. The presence of the cold subsurface waters decreased the mean along-inlet velocities everywhere underneath the surface outflow layer, but particularly at the core of the temperature minimum, where velocities approached zero (Figs. 4 and 5). As mentioned in the introduction, the temperature minimum originated during the previous winter, when a strong Arctic wind outflow event over Bute Inlet vertically mixed the top ~100 m of the water column, leading to cooling and oxygenation down to that depth, effectively creating a new water mass (Jackson et al., 2023). As the surface conditions changed along with the seasons (surface warming in spring/summer as well as freshening due to increased river flow), the new cold water mass became isolated from the surface and remained constrained to the subsurface, leading to the observed profiles used for our initial conditions in May 2019 (Fig. A1). In our simulations, the cold water mass led to higher density and less stratification in the upper ~5 to 50 m of the water column (Fig. 6c,d; Fig. A1), particularly closer to the head of the inlet (Fig. 6b,c). The latter led to a reduced density difference along the fjord near the surface, effectively reducing the strength of the estuarine circulation and decreasing the along-inlet mean velocities (Figs. 4 and 5). The even weaker velocities in the baseline simulation further decreased advection and increased retention in the inlet (Fig. 7), contributing to the ability of the cold water mass to remain in place until external conditions change the dynamics (i.e., the arrival of strong autumn/winter wind-driven deep mixing in addition to the reduced freshwater forcing, which decreases after peaking during the summer).

The mean residual three-layer structure found in Bute Inlet in the absence of the cold subsurface water mass (Figs. 4d, e and 5e to h) was consistent with the expectations for such a deep fjord, following Valle-Levinson et al. (2014) and references therein. These authors propose a dynamical depth, δ, that compares the total water column depth against the depth of frictional

influence in an oscillatory flow ($\delta = \sqrt{\omega H^2 / A_z}$ is the inverse of the Stokes number; $\omega$ is the frequency of tidal forcing, H is the depth of the channel, and $A_z$ is the vertical eddy viscosity). In deep fjords ($\delta > 6$), the frictional depth represents only a small portion of the water column and the tidal residual flow is expected to be three-layered (outflow at the surface and bottom, with inflow in between). Using representative values for Bute Inlet from the model (mean H = 500 m, mean $A_z = 4 \times 10^{-4}$ m$^2$ s$^{-1}$, and $\omega = 1.4 \times 10^{-4}$ rad s$^{-1}$ for the dominant tidal constituent, M2), the resulting $\delta = 296$ is indeed consistent with said three-layer structure.

We recognize the limitations of applying a diffusive numerical framework to a narrow (< 3 km) and deep (~700 m) fjord. However, we note that the flexibility provided by FVCOM's unstructured triangular grid is a significant advantage when modelling areas with many channels and islands, such as the BC region shown in Fig. 1. Current efforts are directed at improving the excess numerical mixing in the model. Nevertheless, we argue that our one-month summer simulation represents observations (particularly density) appropriately and is representative of the late-spring/summer period, since forcing was consistent from mid-May to September (high freshwater discharge and stable wind forcing and open boundary conditions); therefore, the mechanisms described here should be applicable for the whole period.

The results presented here, while specific to Bute Inlet, can be relevant to other fjords in the world. Firstly, we argue that any long, deep inlet even with strong freshwater forcing will have a slow return residual circulation, which could contribute to the persistence of subsurface features inside the fjord. The latter could be particularly relevant if there is a potential source/release of contaminants below the surface outflowing layer. Secondly, we note that katabatic wind events are not specific to Bute Inlet, but have also been observed in other fjords of BC (Pickard, 1961), Alaska (Ladd and Cheng, 2016), southeast Greenland (Oltmanns et al., 2014; Spall et al., 2017), and Antarctica (Forsch et al., 2021). The occurrence of these wind events depends on the location, geography, and topography of the fjord (i.e., a fjord must be connected to the continental plateau to experience these wind events). Thus, our findings from Bute Inlet (regarding the slow circulation that permits a persistent subsurface water mass) could be representative of deep, long fjords in the mid-latitudes that experience katabatic winds or deep-mixing events in winter.

## 6 Summary and conclusions

We described how the geometry of a fjord that experiences strong freshwater forcing at its head can lead to slow residual circulation that allows for subsurface features to persist. Furthermore, if the subsurface feature is a cold (and oxygenated) water mass originated the previous winter due to a katabatic wind event, changes in density can decrease the strength of the estuarine circulation and further contribute to the preservation of the water mass. In the context of climate change, with globally observed oxygen declines and warming temperatures in fjords (e.g., Aksnes et al., 2019; Jackson et al., 2021; Linford et al., 2023), mechanisms that allow for persistent cold and oxygenated waters could alleviate some of the negative consequences of global warming and potentially create temporal ecological refugia.

### Appendix A

Profiles of initial conditions in a mid-inlet station ('BU4' at 50.6ºN and 124.9ºW) help further describe the difference between the baseline and sensitivity simulations (Fig. A1). The figure also shows the observations from 23 May 2019 that were used to create the baseline initial conditions. Lastly, the comparison between modelled and observed time series of surface elevation at the Campbell River tidal gauge provide an additional visualization of the evaluation of model performance (Fig. A2).

### Appendix B

The month of simulation (late May to end of June 2019) is deemed to properly represent the summer conditions in Bute Inlet, given that the river discharge was already high (Fig. B1), the wind blew consistently from the south until September (Fig. B2), and the temperature and salinity at the open boundaries were already representative of the warmer season (Fig. B3).

### Code availability

FVCOM code is publicly available by their developers at https://github.com/FVCOM-GitHub/FVCOM. The Lagrangian
particle tracking model PTrack is publicly available at https://github.com/VA1TRL/PTrack.

### Data availability

Ocean observations used in this manuscript can be found in the publicly available Canadian Integrated Ocean Observing System repository (https://www.cioos.ca/). River discharge data are publicly available through the Water Survey of Canada (https://wateroffice.ec.gc.ca/). Furthermore, both ocean model outputs and observational datasets used for this publication are
accessible from https://doi.org/10.5281/zenodo.10607078 (Bianucci, 2024). Further model data are available upon request.

### Author contribution

LB conceptualized the project, obtained funding, performed model simulations and analysis, produced the figures, wrote the original draft, and led the manuscript to its final form. All co-authors participated in the review and editing of the draft. JMJ and SEA contributed to the analysis and interpretation of model results. MK improved the model grid in Bute and Toba Inlets
and contributed to the finalization of the figures. IJWG improved the methods to estimate river discharge for ungauged rivers and provided the related data. WC contributed the particle tracking experiments.

### Competing interests

The authors declare that they have no conflict of interest.

**Acknowledgments**

This work was funded by Fisheries and Oceans Canada (DFO) partly through the Program in Aquaculture Regulatory Research. The authors are grateful to members of the UBC-DFO Modelling Working Group for many useful discussions and suggestions during our group meetings, including (but not limited to) Mike Foreman, Michael Dunphy, Amber Holdsworth, Jonathan Izett, and Di Wan. Hayley Dosser, Shani Rousseau, Peter Chandler, and Pramod Thupaki provided support at early stages of this work. Di Wan provided an internal DFO review before submission. Two anonymous reviewers and the editor

provided valuable feedback. Observations in 2019 were taken with the collaboration of Homalco First Nation. We gratefully acknowledge that ocean sampling in this work took place on the traditional territories of the Homalco, We Wai Kai, Wei Wai Kum, Kwiakah, and Tla'amin Nations.

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

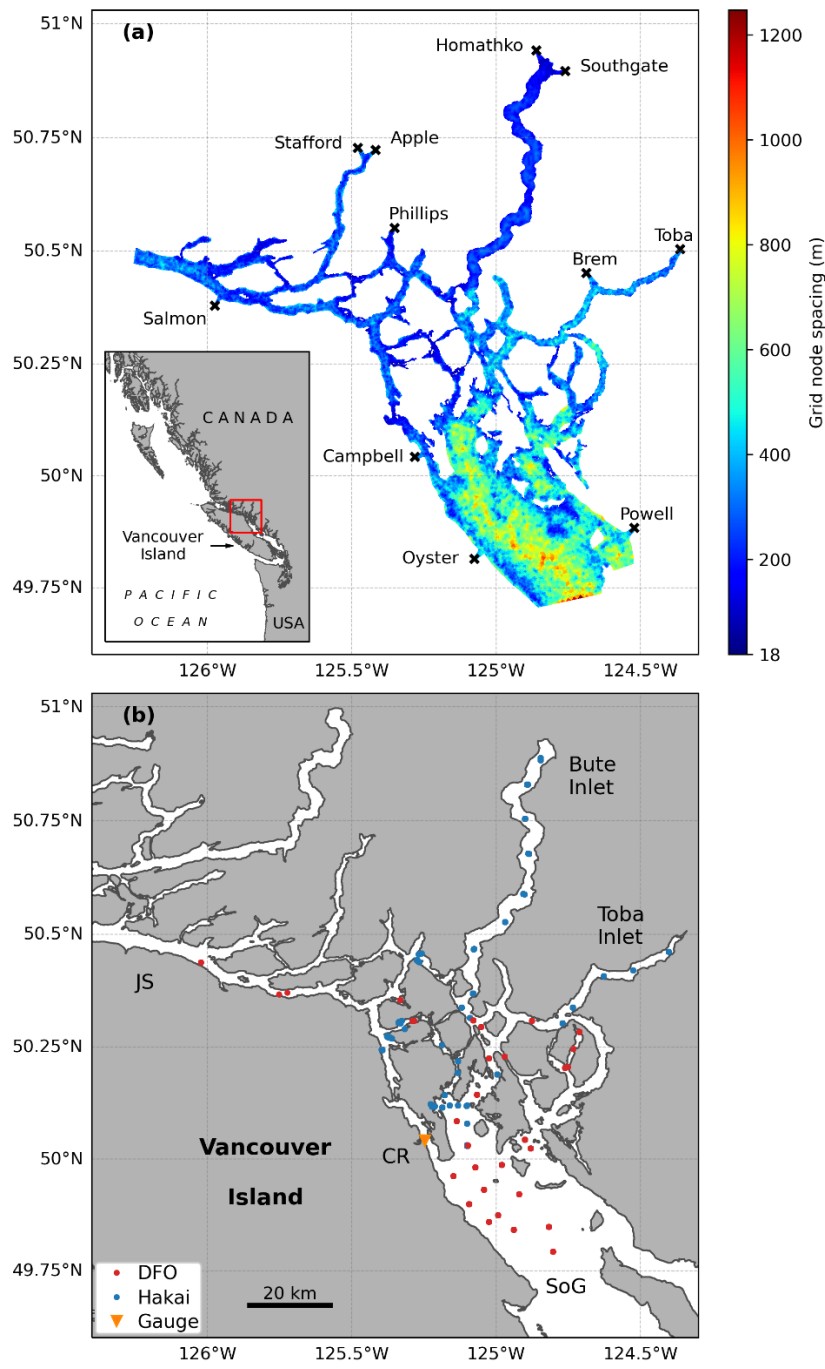

Figure 1: (a) Model grid and location of rivers used to force the model; the red box in the inset shows the domain location on the west coast of Canada. Colours represent the resolution (in meters) calculated as the square root of twice the area of a triangular element. (b) Location of the observations for the period of study, colour coded by source (Fisheries and Oceans Canada/DFO and Hakai Institute); the triangle depicts the location of the Campbell River tidal gauge. Geographical places mentioned in the text are shown: Bute Inlet, Toba Inlet, Strait of Georgia (SoG), Johnstone Strait (JS), Campbell River (CR) and Vancouver Island.


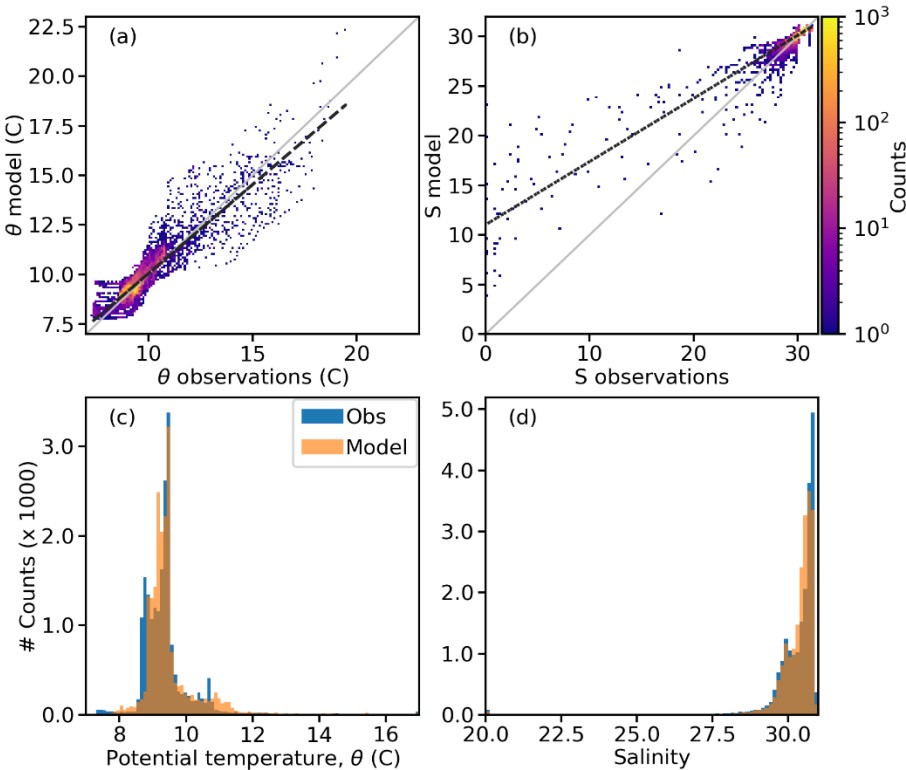

**Figure 2. Two-dimensional histograms for (a) potential temperature and (b) salinity; the grey line shows the 1:1 slope and the dashed black line, the linear regression fit. Overlapped model and observed histograms for (c) potential temperature and (d) salinity.**

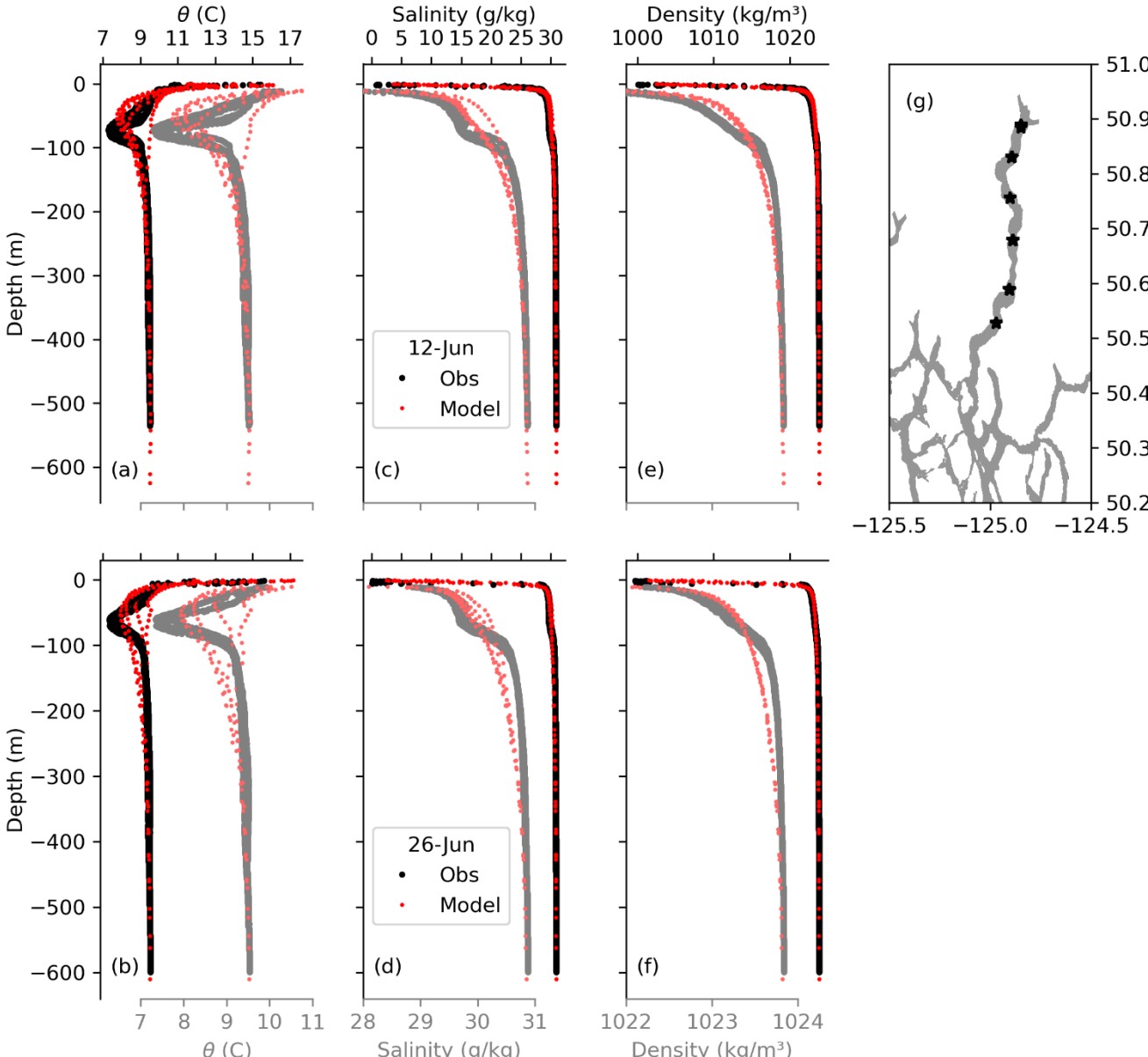

**Figure 3.** Comparison of modelled (red) vs. observed (black) profiles in Bute Inlet. Each row shows profiles for a given date (12 and 26 June 2019). Variables shown are (a, b) potential temperature, (c, d) salinity, and (e, f) density. (g) Locations of the profiles in Bute Inlet. The values below 10 m depth are shown with expanded x-axes in grey and faded red colours, with their corresponding axes at the bottom.

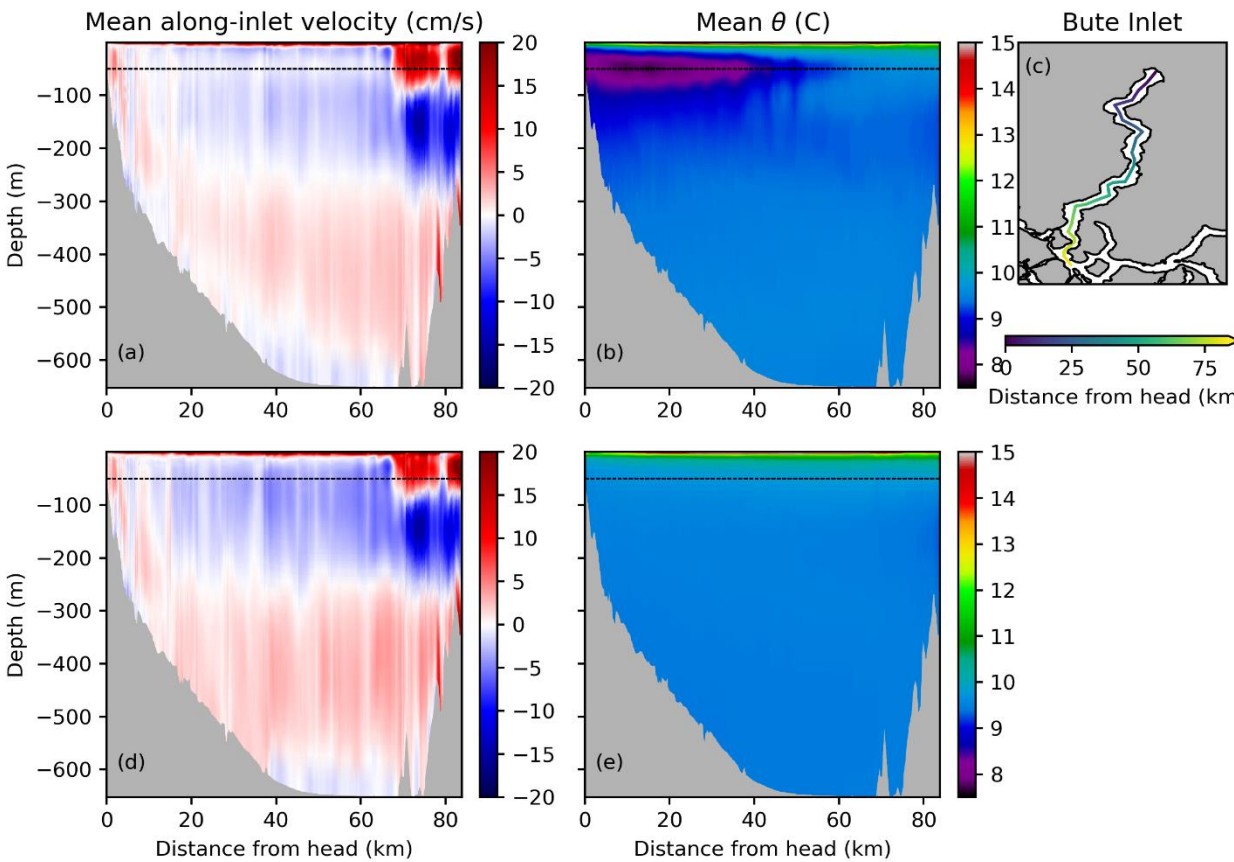

**Figure 4. Mean along-inlet transects throughout Bute Inlet for (a, b) baseline and (d, e) sensitivity simulations. Variables shown are**
**(a, d) mean along inlet velocity and (b, e) mean potential temperature. Velocities are positive (red) towards the mouth of the inlet**
**and negative (blue) towards the head. Averages over the last 29 days of the simulation removed tidal effects. Dotted horizontal line**
**at 50 m highlights the location of the mean temperature minimum in all panels. (c) Map of Bute Inlet transect, colour-coded by the**
**distance from the head of the inlet.**


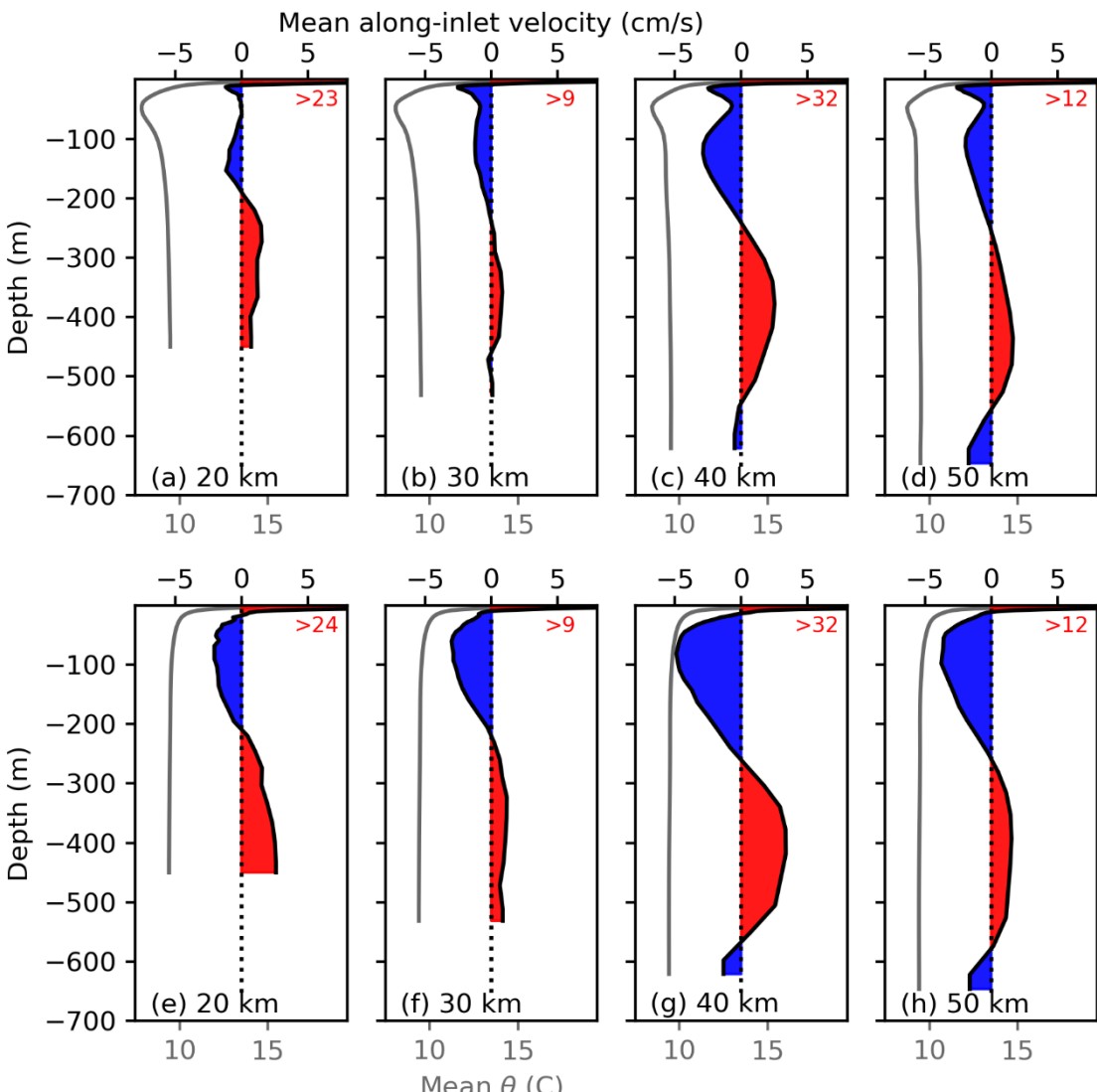

**Figure 5.** Vertical profiles of mean along-inlet velocity (coloured red/blue) and potential temperature (grey) for the (a-d) baseline and (e-h) sensitivity simulations, at four locations in the inlet (from left to right: 20, 30, 40, and 50 km away from the head; see Fig. 4c). Velocities are positive (red) towards the mouth of the inlet and negative (blue) towards the head. Velocity values for the outflowing surface layer are given as red numbers on the top-right of each panels (in cm/s)

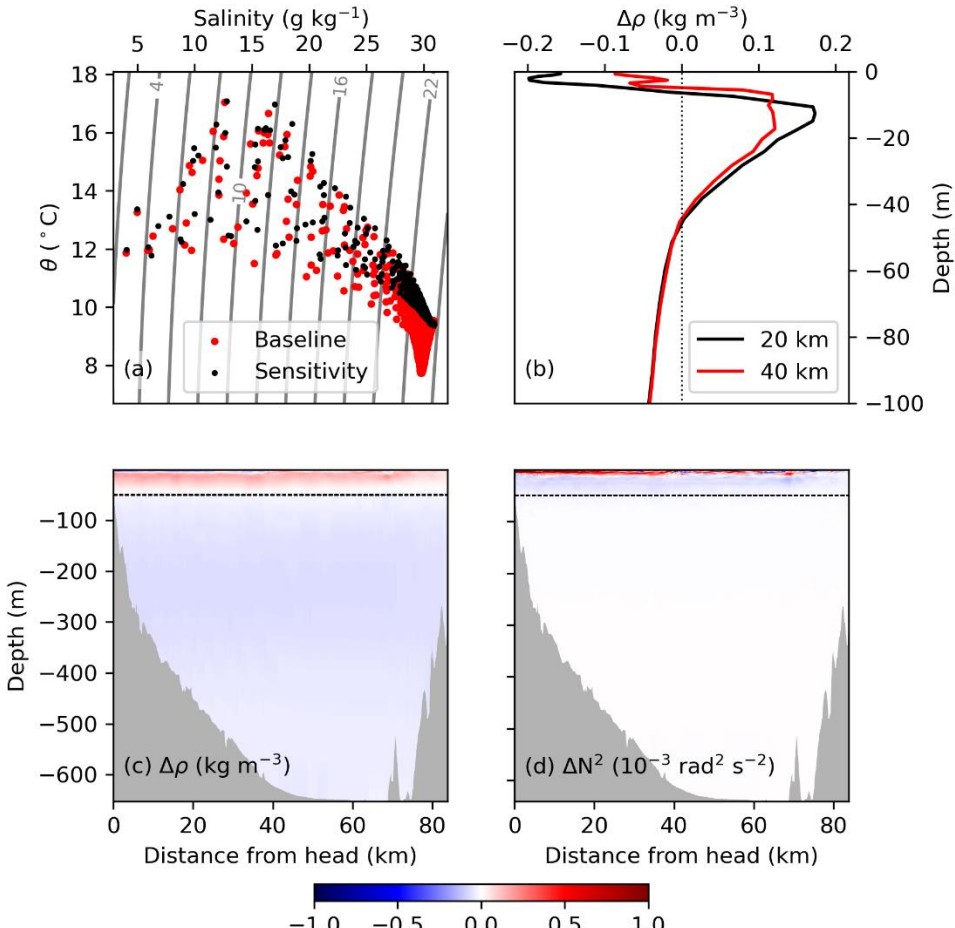

**Figure 6. (a) Temperature-salinity diagram for the baseline (red) and sensitivity (black) simulations in Bute Inlet. Model results shown at the time and location of the observations (12 and 26 June 2019; location in Fig. 4g); for reference, four isopycnals were labelled according to their $\sigma_\theta$ (kg m$^{-3}$). (b) Profiles of mean density difference between of both simulations ($\Delta\rho$) at 20 and 40 km away from the head of the inlet, shown in the top 100 m of the water column. Bottom panels show along-inlet transects of the difference in (c) mean density and (d) mean Brunt-Väisälä frequency (N$^2$) between baseline and sensitivity experiments ($\Delta$ = 29-day average baseline minus 29-day average sensitivity; negative values indicate that the baseline is less dense/stratified than the sensitivity simulation).**

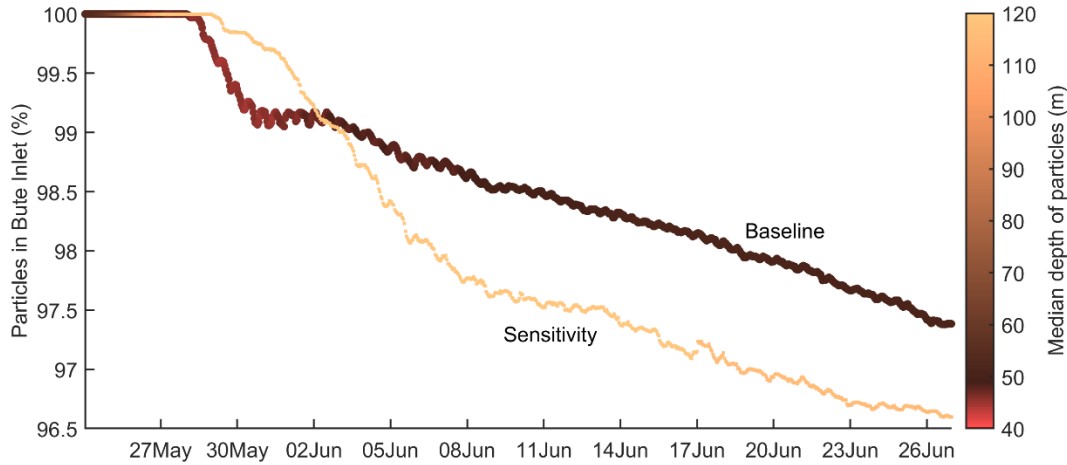


**Figure 7. Time series of total percent of particles retained in Bute Inlet (north of 50.45 °N) in the particle tracking experiments using the velocity fields from the baseline and sensitivity simulations. Lines are coloured according to the median depth of the particles inside the inlet at any given time. Particles were removed from the analysis if and as they reached the coastline or the seafloor.**

**Table 1. Metrics calculated to compare model performance against observations. Potential temperature (θ) and salinity metrics are shown for the whole domain (surface to bottom) as well as the upper 100 m of the water column. N indicates the number of observation-model pairs.**

| Metric | Potential temperature, θ | | Salinity | | Sea level |
|---|---|---|---|---|---|
| | Full domain | Top 100 m | Full domain | Top 100 m | |
| Bias | 0.08 °C | 0.22 °C | 0.05 g kg$^{-1}$ | 0.18 g kg$^{-1}$ | -16 cm |
| RMSE | 0.44 °C | 0.66 °C | 0.73 g kg$^{-1}$ | 1.14 g kg$^{-1}$ | 21 cm |
| Skill | 0.81 | 0.80 | 0.80 | 0.79 | 0.95 |
| Willmott skill | 0.95 | 0.95 | 0.93 | 0.92 | 0.99 |
| R$^2$ | 0.82 | 0.82 | 0.87 | 0.86 | 0.98 |
| N | 20147 | 7818 | 20231 | 7902 | 817 |

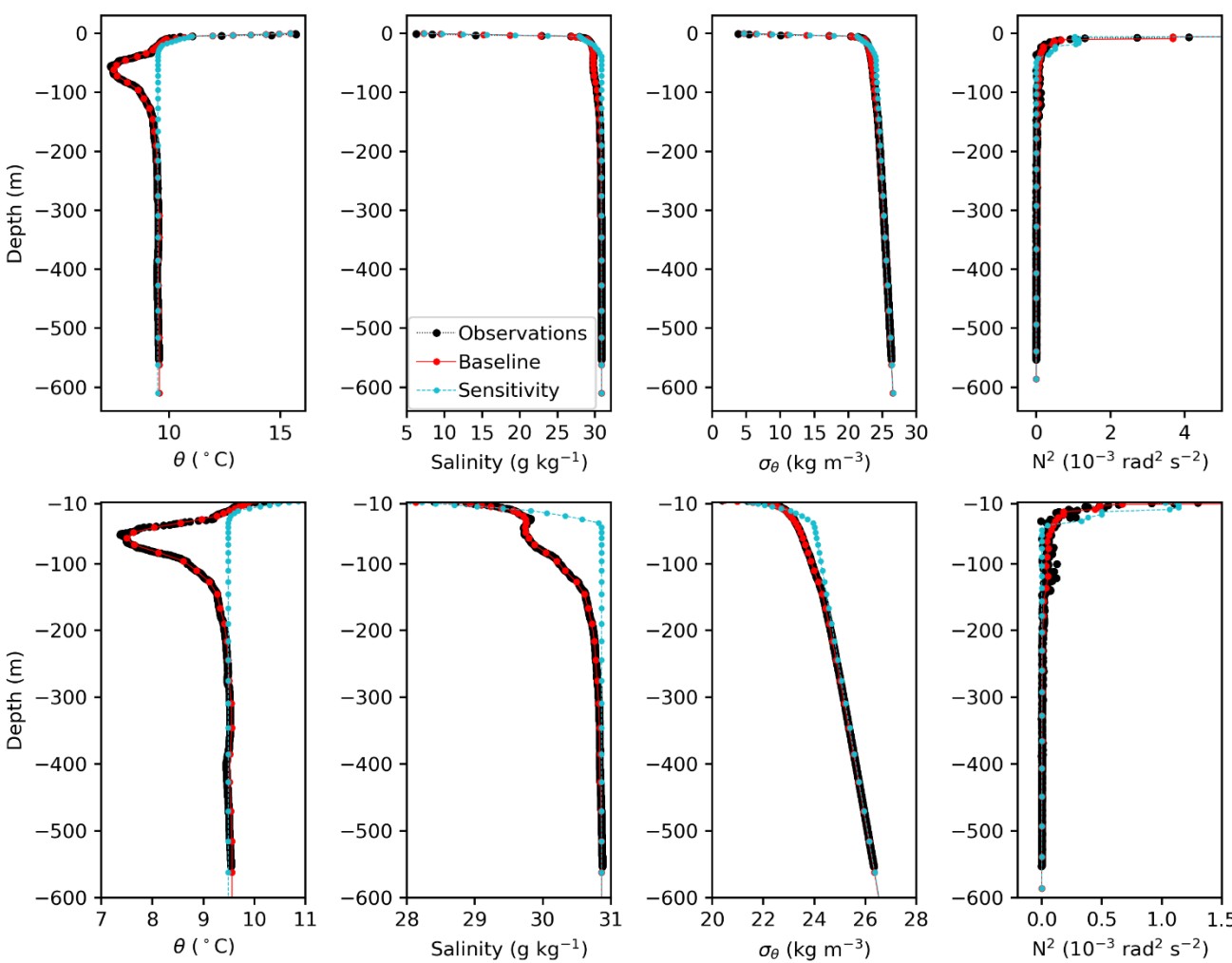


**Figure A1. Profiles of potential temperature, salinity, density (as $\sigma_\theta$), and Brunt-Väisälä frequency ($N^2$) at station BU4 (middle of the inlet at 50.6ºN and 124.9ºW) for the observations on 23 May 2019 (black) and the initial conditions for the baseline (red) and sensitivity (blue) simulations. Top row shows the whole water column and bottom row focuses on values below 10 m.**

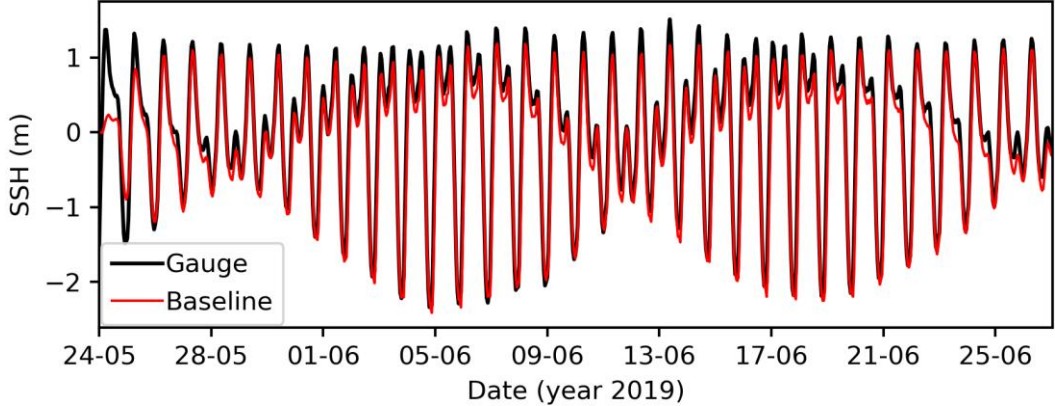


**Figure A2. Observed (black) and modelled (red) time series of sea surface height (SSH) at Campbell River. Location of tidal gauge shown as a triangle in Fig. 1b.**

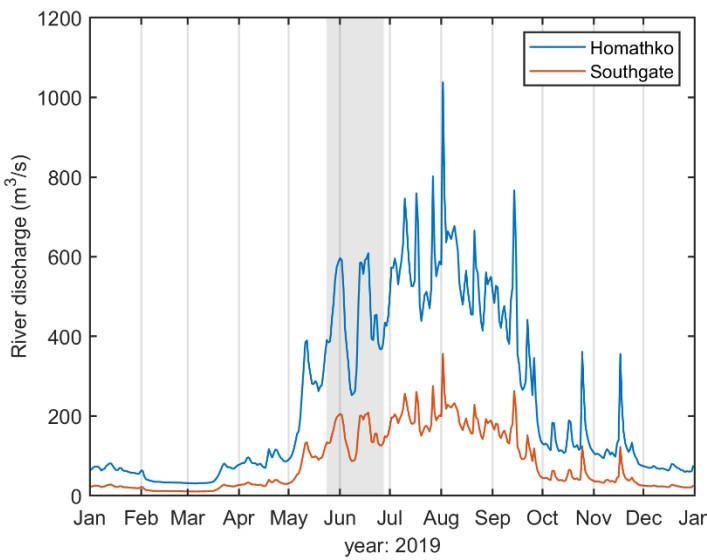

**Figure B1. Time series of freshwater discharge from the two largest watersheds at the head of Bute Inlet: Homathko (gauged) and**
**Southgate Rivers (estimated from Homathko River with a watershed-area ratio, see section 2.2). Discharge is shown for all of 2019 and grey shading indicates the time period of the simulations.**

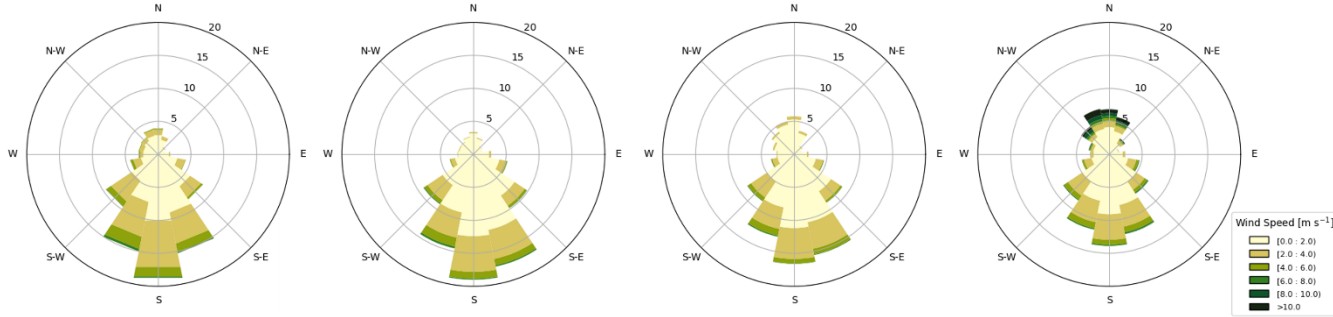

**Figure B2.** Wind roses of the atmospheric forcing over Bute Inlet for the months of June to September (from left to right). Note: the wind rose of June includes the last week of May (from May 24). Colours indicate wind speed; the radius represents the percentage of winds blowing from in a given direction (the latter discretized every 22.5 degrees).

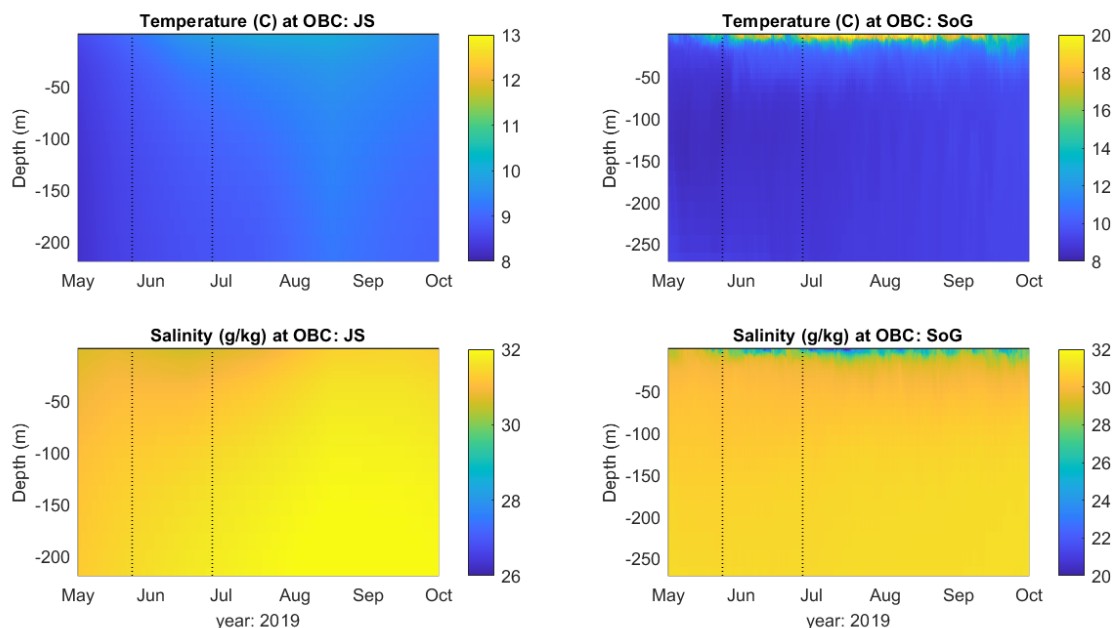

**Figure B3.** Time series of temperature and salinity vertical profiles at one node in each open boundary: Johnstone Strait (JS) and Strait of Georgia (SoG). Vertical dotted lines indicate the time period of the simulations.