# Peer review of "Fjord circulation permits persistent subsurface water mass in a long, deep mid-latitude inlet"

_EGUsphere, 2023_

## Author Response (AR1)

Comments by Reviewer 1 and 2 follow below, including our responses. The original responses to reviewers are in blue and the corresponding modifications to manuscript are in purple.

**Reviewer 1**

The authors use a high-resolution, unstructured model to investigate the persistence of a cold, oxygen-rich sub-surface layer formed during the preceding winter. They show that the presence of the layer – and the stratification changes that it brings about – changes the background circulation from a three-layer to a four-layer system, and suggest that increased mixing at the head of the fjord reduces the estuarine circulation.

The paper is clear and well written – but the scientific argument is relatively weak, and the results could be better quantified and presented. Rather than exploring how the cold anomaly can persist, which is what they set out to do according to the abstract – the paper is a comparison of the circulation within the fjord during a short period in June for experiments with and without the cold layer present. We are shown that the circulation changes – but the authors do not explain why. Mixing is stated to be weaker due to the circulation changes, but this (or the effect on the cold layer) is not shown/quantified. Is the difference in mixing between the two scenarios larger than the difference between the model and the observations? (which is mentioned in the text and, seen in the excessive "smoothing" of the modelled T-profiles in Fig. 3). What is the "normal" residence time for water at one level in the fjord – and how does that change with the "perturbed" stratification?

The model was run for one month – but there is no mention of how the boundary conditions change throughout the period (or the year) and how this would affect the circulation.

In addition, I find that the choice of figures illustrating the points could be improved (see detailed suggestions below for a few suggestions).

I can recommend publication only after major revision.

Please find below the original response to reviewers in blue, and the corresponding modifications to manuscript in purple.

We appreciate Reviewer1's (R1) insightful comments, which we believe will help improve our manuscript. We plan to modify both the title and focus of the manuscript, while improving some descriptions/explanations. In particular, we will now emphasize that the overall sluggish circulation of this long, deep inlet is the main reason why the winter-formed subsurface water mass persists (the slow circulation, typical of long and deep fjords, is seen in both simulations, although it is even slower when the subsurface water mass is present). A new particle tracking analysis shows that >97% of the particles released at the start of the simulations from the location of the temperature minimum layer

(hereafter referred to as "Tmin") remain inside the inlet after 34 days (see more details in response to L228). We will add this analysis to our manuscript, adding a co-author to the paper (Wendy Callendar contributed the particle tracking experiments).

We address R1's general comments regarding circulation changes, mixing, model diffusivity, and residence times in the responses to the specific comments below. We will make sure the text is clear regarding the circulation, which is driven mainly by density. Furthermore, we plan to reduce the focus on the layering of the circulation in the simulation with the subsurface Tmin (baseline simulation) and will emphasize the slower velocities (i.e., advection) as the reason for the persistence of the subsurface feature.  The new title will be "**Fjord circulation permits persistence of subsurface water mass in a long, deep mid-latitude inlet**" and we will adapt the abstract and main text to reflect the subtle change of narrative.

Lastly, regarding R1's comment on the boundary conditions and how they change during the month of simulation: given the transit time analysis (well over a month for waters that inflow into Bute Inlet; see more details in response to L160), it is unlikely that the conditions at the open boundaries (Johnstone Strait and the Strait of Georgia) will affect the fjord during the simulation period. Those open boundary conditions are taken from a larger scale regional model. River forcing (mostly dominated by the Homathko River and followed by the Southgate River) had its most rapid change during May 2019, such that during our simulation period, discharge is already high (reflecting the snow and glacier melt that drives discharge in these watersheds). Wind forcing is also stable (as shown by the wind rose in Fig R1.1), mostly blowing from the south. All of this information will be better explained/added into the methods section of the manuscript.

[Figure]

Fig R1.1: Wind rose of the atmospheric forcing over Bute Inlet. Most of the winds blow from the south. Colours indicate wind speed; the radius represents the percentage of winds in a given direction (the latter discretized every 22.5 degrees).

Addition to the text in section 2.3:

*Nevertheless, the chosen month properly represented the summer conditions in the inlet, since the freshwater forcing was high from late May to late September, the wind conditions were stable (blowing mostly from the south until September), and the values at the open boundaries were similar throughout the summer (see Appendix B).*

Then, Appendix B (please see Figures B1, B2, and B3 in the manuscript):

*The month of simulation (late May to end of June 2019) is deemed to properly represent the summer conditions in Bute Inlet, given that the river discharge was already high (Fig. B1), the wind blew consistently from the south until September (Fig. B2), and the temperature and salinity at the open boundaries were already representative of the warmer season (Fig. B3).*

Specific comments:

L 47: Explain here how this changed the stratification/layering of the fjords (e.g. using text from line 220) - and refer to Fig. A1 (which ought to be included in the main paper). Consider including also a profile from a non-Arctic outflow year in Fig A1.

Thank you for the suggestion, we will better describe the formation of the cold, oxygenated subsurface water mass in the introduction (adding information like that described in line 220 of the original manuscript). However, we do not believe that figure A1 belongs to the introduction section because of two main reasons: 1) this figure has information related to the two simulations, neither of which has yet been introduced at this point of the manuscript; 2) the feature can be easily observed in the cited literature (Jackson et al 2023 and also MacNeill 197 and Pickard 1961), so we argue that a reference to the literature should suffice here. Given our decision to not to refer to Fig A1 here, we are not adding the non-Arctic outflow year to the figure (more on "normal years" in response to comments L222). However, please note that we will add a panel to Fig A1 showing N^2 for the observations and both simulations (shown in Fig R1.6).

Modifications to the manuscript's introduction (new text in *italics*):

Despite the observed warming and deoxygenation at depth, recent observations in Bute Inlet showed that subsurface (i.e., above the sill depth) cold and oxygen-rich waters originated during an Arctic outflow wind event in February 2019 and persisted until the following fall (Jackson et al., 2023). *These authors determined that the strong, cold February katabatic winds mixed the top ~100 m of the water column in Bute Inlet, leading to cooling and oxygenation down to that depth; they also observed temperature minima and oxygen maxima in monthly vertical profiles from March until October 2019.*

L 97: The description of the river-forcing is very detailed – consider moving it to the appendix.

This description will be simplified in the main text (e.g., removing references to specific river gauge IDs)

Wording was simplified and gauge identification numbers were removed. Please look at the third paragraph of section 2.2 of the revised manuscript.

L127: What do you mean by "mostly limited to"

We wanted to convey that we did not have access to many other datasets other than CTD profiles (e.g., we could only evaluate our model against CTD profiles and sea surface elevation from one tidal gauge). We will simplify this sentence to read "Observed vertical profiles were available from both bottle and CTD measurements"

Sentence now reads "*Observed vertical profiles were available from…*". Note this sentence was moved to section 2.4, which was re-structured slightly and now is called "Available observations and metrics for model evaluation" (instead of "Metrics for model evaluation")

145 and on: How useful are these metrics - as used here - when the larger part of the water column does not change during the short simulation, i.e. it's all about the initial conditions? In addition, there are now observations of velocity, on which the paper's main results are based.

We could certainly provide the model evaluation for the top 100 m of the water column, which would not be as affected by the bottom conditions (that do not change a lot during the month of simulation). We include below the evaluation (plots and metrics) for the upper 100 m and compare those against the plots and calculations for the whole water column; the overall performance does not change significantly if we only focus on the upper waters. The bias and RMSE are larger in the 100 m metrics, partly due to the numerical diffusivity issues discussed in the next response (L158). We argue that the qualitative (plots) and quantitative (metrics) evaluation of the model performance is an important component of a modelling paper. Therefore, we believe we should keep these metrics. We plan to show the plots for the whole domain evaluation, but add text to refer to the top 100 m evaluation (unless the editor/reviewers feel strongly that we should take the opposite approach, and show the 100 m evaluation and only discuss the full domain metrics).

Lastly, we know that instruments to measure currents in Bute Inlet are going into the water in 2024 (by The Hakai Institute), but are not yet deployed; therefore, we are not sure about the observations of velocity that R1 is referring to. The only current observations we found were from downward-looking ADCPs that focused on near-bottom currents in 2018 (for the study of turbidity currents, see Lewis et al, 2023), so not meaningful to evaluate the model results on the upper part of the water column that are the focus of this paper.

Reference:

Lewis P.B., M.A. Clare, E.L. Pope, I.D. Haigh, M.J.B. Cartigny, P.J. Talling, D.G. Lintern, S. Hage, M. Heijnen. Predicting turbidity current activity offshore from meltwater-fed river deltas, Earth and Planetary Science Letters 604. Doi: 10.1016/j.epsl.2022.117977, 2023.

[Figure]

Model evaluation for the whole water column
(Fig 2 in manuscript)

Model evaluation for the upper 100 m
(analogous to Fig 2, but only for top 100 m)

Table R1.1: comparison of metrics for the full domain (Table 1 in manuscript) vs. top 100 m.

| Metric | Potential temperature, θ | | Salinity | |
|---|---|---|---|---|
| | Full domain | Upper 100 m | Full domain | Upper 100 m |
| Bias | 0.08 °C | 0.22 °C | 0.05 g kg$^{-1}$ | 0.18 g kg$^{-1}$ |
| RMSE | 0.44 °C | 0.66 °C | 0.73 g kg$^{-1}$ | 1.14 g kg$^{-1}$ |
| Skill | 0.81 | 0.80 | 0.80 | 0.79 |
| Willmott skill | 0.95 | 0.95 | 0.93 | 0.92 |
| R$^2$ | 0.82 | 0.82 | 0.87 | 0.86 |
| N | 20147 | 7818 | 20231 | 7902 |

Modifications to the manuscript (new or modified text in *italics*):

End of new first paragraph of section 2.4: *Unfortunately, no observed velocity profiles were available in Bute Inlet to evaluate the modelled currents.*

Second paragraph of section 2.4: With all the available pairs, we calculated several metrics frequently used to quantify model-observations misfit *(both for the full model domain and for the top 100 m only)*.

Section 3: The model performance was evaluated through both quantitative and qualitative approaches. The quantitative metrics showed a good agreement between model and observations, *both for the whole model domain and for the upper 100 m* (Table 1). *For the whole water column,* biases were less than one tenth for both temperature and salinity (0.08 °C and 0.05 g kg$^{-1}$, respectively), while RMSEs were below 0.5 °C and 0.8 g kg$^{-1}$. *Bias and RMSE values were somewhat larger if only the top 100 m were considered (Table 1), given the larger range of conditions in the upper layers; for instance, the model has trouble representing the fresh/brackish waters in Bute Inlet*

*(Fig. 2b)*. All non-dimensional metrics were at or above 0.8, with particularly high Willmott skill scores *above 0.92 for both* temperature and salinity.

Table 1: columns for upper 100 m added for temperature and salinity; caption modified accordingly.

158 and on: Nor sure I agree - if you zoom in on the cold, subsurface layer and use a scale that's adapted, there are quite some differences in the depth, "sharpness" and the vertical extent of the cold layer already a fortnight after model initialization. I'd suggest plotting the profiles of temperatures from the three occasions on top of each other (modeled in one panel and observed in one) to show the time development in the model vs. obs. If one assumes advection to be negligible, one can (I think?) use the differences in the profiles to infer an estimate of observed/modeled diffusivity.

We did not intend to 'over sell' the ability of the model to represent the vertical structure of the observations. We will improve the wording to make sure the model performance is not overstated. We did mention in Line 160 of the original manuscript that the model did not show the temperature minimum as sharply as the observations, due to numerical diffusion. It is a very common problem for diffusive models (such as FVCOM) to represent maxima/minima features.

We show below the figure suggested by R1 (Fig R1.2), with observed and modelled profiles of T and S at the station BU6 (in the middle of the inlet, see inset) in June 12 and June 26 (timing of the observations available for model evaluation). The observed profiles do not show much mixing, since the Tmin basically maintains the same value (difference of Tmin is 0.01°C). While the difference in Tmin between both dates is larger in the baseline simulation (0.15°C), it is not many orders of magnitude larger; the same is true (with even smaller differences) for salinity (ΔS at the location of the Tmin for the observations is 0.04 and for the baseline, 0.03 g/kg). Furthermore, below 300 m the observations show small changes in T and S (deltas < or ~0.01 for both) in the 14-day period. The model shows a bit larger differences with both ΔT and ΔS around or less than -0.06 near the bottom. In a month (30d), these numbers could indicate a change in S & T of around -0.13 units.

While the model is diffusive, we argue that the main density driven circulation is still well represented in a month-long simulation. Fig R1.3 shows that the pycnocline is well represented (see more density discussion in the response to comments for Fig 3, particularly for the representation of density at the depth of Tmin). Numerical diffusion could become an issue for longer simulations if it took over the main density structure.

We note the challenges of modelling a fjord that is ~2km wide and ~700m deep using a modelling framework that's typically diffusive. However, we argue that despite this drawback FVCOM still has advantages that make it an appropriate choice for the modelling of fjord systems (e.g., the flexibility provided by the unstructured triangular grid to represent a region replete of channels and islands). We will add a discussion of the advantages and disadvantages of FVCOM in section 5 of the revised manuscript.

[Figure]

Fig R1.2: Observed and modelled profiles of (top) temperature and (bottom) salinity, in June 12 and 26 in station BU6 (mid-inlet, shown in inset). Also showing the difference (Δ) between profiles in the two dates, for both observations and model (baseline simulation that has Tmin in initial conditions).

[Figure]

Fig R1.3: Observed (black) and modelled (red) density profiles at three stations (BU8, BU6 and B4 – see inset for location) for (top) June 12 and (bottom) June 26.

Modifications to the 3rd paragraph of section 3 (new/modified text in *italics*; removed text ):

** The observed temperature profiles *in Bute Inlet in June 12 and 26* showed a temperature minimum around 80 m depth, which was also present in the model results*, albeit somewhat shallower (~45 m) and not as sharply defined* (Fig. 3a, b). *The latter is likely due in part to numerical mixing, since horizontal diffusion in FVCOM occurs parallel to the sigma layers (Chen et al., 2006), a simplification that can lead to an overly diffusive model in regions with steep topography and significant slopes in the terrain-following layers (Foreman et al., 2023). At the location of the observed temperature minimum, salinity and density showed distinct vertical gradients (Fig. 3c-f); these also were diffused in the model. However, the observed temperature and salinity features compensated each other in density, such that overall, the model was better able to represent the density structure at this depth (Fig. 3e, f).* As mentioned before, the model overestimated surface salinity (by several g kg$^{-1}$; Fig. 3c, d), leading also to an overestimation of surface density (Fig. 3e, f). Nevertheless, both the *main* halocline and pycnocline were correctly represented by the model, with a sharp vertical gradient in the top *20* m of the water column. Bottom values were homogeneous and matched the observations below ~*300* m.

New Figure 3, showing both the full water column and everything below 10 m with different x axes:

[Figure]

Figure 3. Comparison of modelled (red) vs. observed (black) profiles in Bute Inlet. Each row shows profiles for a given date (12 and 26 June 2019). Variables shown are (a, b) potential temperature, (c, d) salinity, and (e, f) density. (g) Locations of the profiles in Bute Inlet. *The values below 10 m depth are shown with expanded x-axes in grey and faded red colours, with their corresponding axes at the bottom.*

160. Well this is not very surprising, you initialized the model two weeks earlier with the cold layer present. How long is your run compared to the normal residence time of water in the fjord?

We do not have information on the residence time in the fjord, but have calculated the transit time with our model results (i.e. length of Bute Inlet divided by the mean velocity in Bute). At the depth range of the Tmin in the baseline simulation (40-50 m), the transit time is over 100 days. Furthermore, in the new particle tracking analysis (see the response to L228) we show that over 97% of particles released at the location of the Tmin layer stay within the fjord after the end of the simulations. Therefore, we should definitely expect the model to have a Tmin in June 12 and 26 (19 and 34 days after initialization, respectively).

L 170 It is not easy to see the structural difference between Fig. 4 a and d that you describe in the text. To me there're four layers in both of the figures: red, blue, red, blue – but I understand that what you refer to as four layers are red, blue, blue red, where the two blue layers are separated by white?

We will clarify and improve the description if Figure 4a,d in the text. We had previously ignored the very bottom blue layer, which we will describe when revising of the manuscript. We added a horizontal dashed line at 50 m, to highlight the (co-)location of the Tmin and the region with almost zero velocities (Fig R1.4). Please note that we will not be focusing as much on the layering in the new version of the manuscript (although we will mention it and discuss it).

[Figure]

New Figure 4 caption: Mean along-inlet transects throughout Bute Inlet for (a, b) baseline and (d, e) sensitivity simulations. Variables shown are (a, d) mean along inlet velocity and (b, e) mean potential temperature. Velocities are positive (red) towards the mouth of the inlet and negative (blue) towards the head. Averages over the last 29 days of the simulation removed tidal effects. *Dotted horizontal line at 50 m highlights the location of the mean temperature minimum in all panels.* (c) Map of Bute Inlet transect, colour-coded by the distance from the head of the inlet.

Modified text in section 4.1 (modifications in *italics*; removed text not shown here for clarity):

A transect plot of mean along-inlet velocities through Bute Inlet showed a *multi*-layered structure of the velocity field in most of the fjord (Fig. 4a). The surface layer flowed outwards of the fjord, with a return flow underneath *down to approximately the depth of the outside sill (~300 m)*, following a typical estuarine circulation. *However, the return flow had a clear vertical structure, with velocities close to zero at the depth of the minimum averaged temperature (Fig. 4a and b; a dotted horizontal line at 50 m highlights the co-location of the near-zero averaged velocities and the mean temperature minimum).* Below the depth of the *outside* sill, the mean, slow flow was towards the mouth of the inlet*, with a narrow and weak inflow layer near the seafloor*.

173. How can there be a net along fjord circulation below sill depth?

We will improve the wording, since outflow "below the sill" sounds counter intuitive. However, it is a feature of the model circulation that the mean direction of the flow below ~280m (approximately the depth of the sill) is towards the mouth of the inlet; this is also seen in the model's across-inlet transects. Observational studies have shown this same feature in other inlets (e.g., Baker and Pond 1995, Castillo et al. 2012, Wan et al 2017) and it was consistent with the expectations for such a deep fjord, following the δ analysis (Valle-Levinson et al., 2014) discussed in section 5.

References:

Manuel I. C, O. Pizarro, U. Cifuentes, N. Ramirez, L. Djurfeldt. Subtidal dynamics in a deep fjord of southern Chile, Continental Shelf Research 49, 73-89, doi:10.1016/j.csr.2012.09.007, 2012

Baker, P. and Pond, S.: The Low-Frequency Residual Circulation in Knight Inlet, British Columbia, J. Phys. Oceanogr., 25, 747–763, 1995

Valle-Levinson, A., Caceres, M. A., and Pizarro, O.: Variations of tidally driven three-layer residual circulation in fjords, Ocean Dynamics, 64, 459–469, doi:10.1007/s10236-014-0694-9, 2014.

Wan, D., Hannah, C. G., Foreman, M. G. G., and Dosso, S.: Subtidal circulation in a deep-silled fjord: Douglas Channel, British Columbia: SUBTIDAL CIRCULATION IN DOUGLAS CHANNEL, J. Geophys. Res. Oceans, 122, 4163–4182, doi:10.1002/2016JC012022, 2017.

Modifications to the text: We changed "Below the sill depth" to "Below the depth of the *outside* sill", to highlight that the sill is quite far (right at the end of the transect shown in Fig 4c, outside of the fjord)

185 I presume that this is because the effect of the salinity changes on the density is greater than the effect of temp. changes (Please quantify) – they would also have different signs, right? If you lower the temperature you make the water at that level denser, while if you make it fresher you make it lighter.

Indeed, density in this region is dominated by salinity (a beta ocean, as in Carmack, 2007). This is clearly seen in Fig 3 and Fig A1, since the shape of the density profiles match very well the shape of the salinity profiles (both in model and observations). A Tmin would not be possible if temperature dominated stratification. It is interesting to note that the upper ~50 m of the water column (from the surface to the core of the Tmin) is double-stable, since temperature decreases with depth while salinity increases. Below the Tmin, where T increases with depth, stratification is solely driven by salinity. Further quantification was done by calculating the density ratio Rρ ($\alpha\frac{\partial T}{\partial z}/\beta\frac{\partial S}{\partial z}$), but we do not feel it adds too much to the analysis. We will emphasize the role of salinity in stratification in the revised version of the manuscript.

Reference:

Carmack, E.C., The alpha/beta ocean distinction: A perspective on freshwater fluxes, convection, nutrients and productivity in high-latitude seas, Deep Sea Research Part II: Topical Studies in Oceanography 54, 2578-2598, doi:10.1016/j.dsr2.2007.08.018, 2007.

Added sentence at the end of section 3: *The strong resemblance of the main halocline and pycnocline (both in the observations and the model) highlight the dominant role of salinity in the stratification of the region (i.e., a beta ocean; Carmack, 2007); clearly, a subsurface temperature minimum is only possible if salinity drives density.*

193 (give depth intervals)

We will specify that the stratification decreased below the outward-flowing layer, between ~5 and 50 m deep.

The information was added to the manuscript: In particular, stratification decreased below the outward-flowing layer *between ~5 and 50 m*

217 Why/how did the temperature minimum create a separation of the return flow?

This statement was poorly written and will be rephrased. The point we will make in the new version of the manuscript is that velocity is slower in the baseline simulation because surface density increased – and it increased more closer to the head than towards the mouth; therefore, the horizontal pressure gradient decreased, slowing down the circulation (see more details in response to L226). The specific density structure leads to a subsurface layer with velocities close to zero at the depth of Tmin. The vertical pressure gradients let us infer that velocities will be smaller at the depth of Tmin, but in such analysis, the zero value would depend on the pressure reference level chosen. Please note that we plan to reduce the focus on the layering of the baseline circulation and will emphasize the overall slow velocities as the reason for the persistence of the Tmin feature.

This specific sentence was modified and now reads: The presence of the cold subsurface waters decreased the mean along-inlet velocities everywhere underneath the surface outflow layer, but particularly at the core of the temperature minimum, where velocities approached zero (Figs. 4 and 5).

291 "velocities were weaker" – please quantify (and or make figures where the reader can directly compare the velocity profiles)

We have improved Fig 5 (following some of the advice from R1) such that the velocities x-axis are zoomed in to better represent the velocities below 10 m (i.e., now the limits for the axis are +/- 8 cm/s instead of going up to 35 cm/s). The surface outflow layer velocities are indicated by a red value near the top of each panel. We have also added a dashed horizontal line to highlight 50m depth (approximate depth of Tmin). Please see below the new Fig 5 and its caption (new text in *italics*).

[Figure]

Fig R1.5: New Figure 5. Vertical profiles of mean along-inlet velocity (coloured red/blue) and potential temperature (grey) for the (a-d) baseline and (e-h) sensitivity simulations, at four locations in the inlet (from left to right: 20, 30, 40, and 50 km away from the head; see Fig. 4c). Velocities are positive (red) towards the mouth of the inlet and negative (blue) towards the head. *Velocity values for the outflowing surface layer are given as red numbers on the top-right of each panels (in cm s$^{-1}$). Horizontal dashed lines highlight 50 m, the approximate location of the temperature minimum in the baseline simulation.*

Figure 5 and its caption were updated in the new manuscript.

222 "salinity and density were impacted" – please describe how (and quantify) – e.g. referring to a new version of Fig A1 where a profile from a "normal year" is included. As of now, the reader has no means to judge whether the salinity profile used in the sensitivity profile (which determines the density and hence the stratification) is realistic.

We thank the reviewer for this question, that made us re-think some of our wording related to the description of the sensitivity simulation. Reviewing the literature (MacNeill 1974 and Pickard 1961) and all available profiles of temperature in Bute Inlet in publicly accessible datasets (both at https://waterproperties.ca and https://cioos.ca), we confirmed that a Tmin is present in Bute Inlet mostly every spring/summer, even if the feature is very small some years (i.e., Tmin only a few tenth of a degree colder than deeper temperatures) and quite large some others (differences well over 1°C, even up to several degrees – see Fig 3 in MacNeill, 1974). Arctic outflow winds are common in BC's winters; Jackson et al (2023) estimated the number of outflow events per decade at Bute Inlet to be between 93 (for the coldest decade, the 1950s) and 20 (for the warmest decade, the 2010s). These numbers imply an average of 2+ outflow events per year. Therefore, even a "normal year" might still have evidence of Arctic outflow events and some kind of Tmin feature. Our sensitivity experiment represents an extreme scenario of a winter without deep mixing, which may not be a "normal year". We argue that it is worth comparing our baseline simulation against the sensitivity experiment (a normal year will be somewhere in the middle), but we will improve descriptions to clarify all of this information in the new version of the manuscript. We can certainly quantify the changes in salinity and density between our two simulations by calculating deltas, although we think those changes are qualitatively displayed in Fig 6a and Fig A1 (we will add references to those figures; Fig A1 will now have a second row, removing the surface values, such that below-surface changes are more easily appreciated – see Fig R1.6 below).


Modifications at old line 222 (re. "salinity and density were impacted"): we removed this sentence and focus on density (rather than salinity) to keep the discussion clearer.

225 Rather than referring to Fig. 6c, refer to a new version of Fig A1 which also includes a panel comparing the initial N2-profile

We will add a reference to a modified Fig A1, which includes a fourth column that shows the initial $N^2$ profiles for the observations, baseline, and sensitivity simulations. It also has now a second row that focuses on the below-surface values:

[Figure]

Fig R1.6: New Figure A1.  Profiles of potential temperature, salinity, density (as σ_θ), and Brunt-Väisälä frequency ($N^2$) at station BU4 (middle of the inlet at 50.6°N and 124.9°W) for the observations on 23 May 2019 (black) and the initial conditions for the baseline (red) and sensitivity (blue) simulations. Top row shows the whole water column and bottom row focuses on values below 10 m.

Modified text in *italics*:

*As* the surface conditions changed along with the seasons (surface warming in spring/summer as well as freshening due to summer *increased* river *flow*), the new cold water mass became isolated from the surface and remained constrained to the subsurface, *leading to the observed profiles used for our initial conditions in May 2019 (Fig. A1). In our simulations, the* cold water mass led to higher density and less stratification in the upper ~5 to 50 m of the water column (Fig. 6*c,d; Fig. A1*), particularly closer to the head of the inlet (Fig. 6b,*c*).

226 Give depth range

We will add the depth range of the denser upper waters  (~5 to 50 m)

This information was added to the text.

226: Please include figures/numbers that show the reduced density difference/estuarine circulation.

We will add references to Fig 5, which will now have a few changes to make it easier to see the difference in the circulation of both simulations (see new version of Fig 5 shown here as fig R1.5). The difference in density will be highlighted by referring to Fig 6b and adding a new figure (or likely, new panel in Fig 6) that shows the profile of Δρ in 2 locations along the inlet. An example of such a figure/panel is shown below (Fig R1.7):

[Figure]

Fig R1.7: New figure or new panel in Fig. 6. Profiles of difference in mean density between baseline and sensitivity experiments at three locations in the fjord: 20 and 40 km away from the head (Δρ = 29-day average baseline minus 29-day average sensitivity; positive values indicate that the baseline is denser than the sensitivity simulation).

Below the surface outflow, between ~5-50m, the baseline simulation is denser. Also, the density increase in the baseline simulation is higher nearer the head of the fjord (20km, black line) than closer to the mouth (40km, red).

References to Figures 4, 5 and the new panel in figure 6 (currently labelled 6b) were added.

L228: "decreased mixing". What is this statement based on? Fig 6b shows that the density decreased all the way to the bottom for the baseline exp? I presume / guess / hope that the difference below e.g. 250 m between the simulations is zero at the start of the simulations.

This sentence will be reformulated and improved. First of all, we are focusing on the top 100 m of the water column, where the Tmin is found. The weaker velocities at those depths (particularly around 50 m) in the baseline simulation lead to decreased advection. Regarding "decrease mixing" we were referring to the smaller horizontal mixing eddy parameters that we found in the baseline simulation (up to 25% or ~1 m$^2$/s smaller in the upper 100 m, figure not shown). However, we realized that those changes are quite small and likely not a key player for allowing the permanence of the Tmin water mass at the subsurface of the inlet. Furthermore, we did a new particle tracking analysis that emphasizes the dominant role of advection. We used weightless virtual particles that flow with the 3D current field – these particles are at all times neutrally buoyant (i.e., no sinking towards the bottom nor rising towards the surface). We deployed these particles at the start of the simulation in the location of the Tmin (i.e., in every model node and level with T<8°C inside Bute Inlet in the initial conditions). By the end of the simulation, over 97% of the particles are still within Bute Inlet in both simulations (see Fig R1.8).

Therefore, we will remove the reference to mixing in this sentence, emphasizing the role of advection in the persistence of the Tmin feature. We will also add the new particle analysis to the manuscript.

[Figure]

Fig R1.8. Time series of total percentage of virtual particles inside of Bute Inlet in both simulations. The number of particles and their initial location is the same in both cases. Final values are 97.1 and 97.6% for the baseline and sensitivity simulations, respectively.

Regarding R1's presumption: As described in the methods section, the initial temperature and salinity profiles of the sensitivity simulation in Bute Inlet were constant below the main pycnocline; the constant values were selected as the coolest and saltiest observations in the deepest third of the water column. Therefore, the densities below 250 m are not exactly the same between simulations, but they are very close (and basically zero below 400 m. This can be seen in the second row of the new Fig A1 (Fig R1.6), since the red and cyan dots are really close to each other, but not exactly on top of each other. However, we argue that the differences are small enough and away from the focus of this work. The reasoning behind the setup of our sensitivity experiment was to avoid the introduction of strange vertical gradients from the homogenization of just a layer of the water column (so we homogenized the whole water column below the pycnocline).

We have removed the mention of mixing and have added the particle tracking experiment showing high retention. The figure above included all particles, even those that grounded on the coastline or seafloor. The new Fig 7 (see below) removes grounded particles and shows even higher retention for the baseline experiment. Additions related to the particle tracking experiments/analysis:

[revised manuscript text omitted]

L 240 responds? I do not understand this sentence.

This sentence will be rewritten, given that we no longer consider the sensitivity experiment to represent "standard summer conditions". The original sentence should have read "shows" instead of "responds to".

Sentence removed, given the change of focus (no longer on the layering of the residual circulation)

L242 give depth range

Will do (it's ~5 – 280m)

Same as above

L 245 Is it relevant to mention deep water renewals here?

We agree that this information should be moved to the discussion (it does not belong to the summary/conclusions)

Information removed from the conclusions

L 249 Do you think/mean that the findings from Bute inlet is universal? Would it not depend on the stratification outside of the fjord? What are the "mechanisms" that you refer to?

There are two separate aspects to consider regarding the universality of our Bute Inlet results. First, there is the presence of Arctic outflow events (also known as gap or katabatic winds) that can create a subsurface water mass in winter. The presence of these wind events depends on the location, geography, and topography of the fjord (i.e., a fjord must be connected to the continental plateau to experience these wind events). Several inlets experience Arctic outflow events in the west of Canada (Pickard 1961) and katabatic winds have been studied in other regions such as Alaska (Ladd & Cheng, 2016), southeast Greenland (Oltmanns et al., 2014; Spall et al., 2017), and Antarctica (Forsch et al, 2021). The second aspect when considering the universality of our Bute Inlet results is the geometry of the fjord itself (i.e., length, depth, width). The freshwater forcing in these fjords implies S=0 at the head and some oceanic S at the mouth (~33-35); thus, the horizontal pressure gradient mostly depends on the length of the inlet, such that long inlets tend to have slow circulation below the surface outflow. We think that the findings from Bute Inlet (regarding the slow circulation that allows for subsurface features to stay in place for many months) could be representative of deep, long fjords in the mid-latitudes that experience katabatic winds or winter deep-mixing events. We will add these thoughts into the revised manuscript.

We argue that while the stratification outside the fjord might play a role in the details of the circulation, the key driver is the freshwater forcing leading to the estuarine circulation. As long as there is a freshwater source large enough to create a strong estuarine circulation in a long, deep

fjord, then a subsurface Tmin feature will be able to persist (if such a water mass develops during the winter) from spring to the following autumn.

We will clarify that by "mechanisms", we mean the slow circulation in a deep, long fjord, that allows for the persistence of a subsurface water mass. A positive feedback would provide a secondary mechanism: the presence of a subsurface Tmin leads to an even slower circulation in the fjord at those depths, which contributes to the decreased advection of the cold waters and, in a small degree, the persistence of the subsurface water mass in the fjord.

The discussion section was modified substantially to add/clarify all of the information above (new text in *italics*; deleted text not shown for clarity):

Section 5. Paragraph 1 ("mechanism #1"): *The slow circulation in Bute Inlet is partly due to the length of the inlet, since longer distances decrease the pressure gradient between the fresh head of the inlet and the saltier mouth. Furthermore, the significant depth of the sill (~300 m) also contributes to a slow return flow, given the large associated cross-inlet area available to compensate the surface volume outflow. The slow residual velocities below the surface lead to low advection, long transit times, and an overall high retention of particles seen in our (summer) model simulations. Therefore, we identify the geometry of a long, deep inlet with freshwater forcing at its head as a main mechanism leading to the long persistence of a subsurface feature.*

Paragraph 2: *A second mechanism is a positive feedback related with the existence of the subsurface water mass.* The presence of the *cold subsurface waters decreased the mean along-inlet velocities everywhere underneath the surface outflow layer, but particularly at the core of the temperature minimum, where velocities approached zero* (Figs. 4 and 5). […] *In our simulations, the* cold water mass led to *higher density and* less stratification *in the upper ~5 to 50 m of the water column* (Fig. 6c,d; Fig. A1), particularly closer to the head of the inlet (Fig. 6b,c). The latter led to a reduced density difference along the fjord near the surface, effectively reducing the strength of the estuarine circulation and decreasing the along-inlet mean velocities (*Figs. 4 and 5*). The *even* weaker velocities in the baseline simulation further decreased advection *and increased retention in the inlet (Fig. 7),*

*contributing to the ability of the cold water mass to remain in place until external conditions change the dynamics (i.e., the arrival of strong autumn/winter wind-driven deep mixing in addition to the reduced freshwater forcing, which decreases after peaking during the summer).*

Last paragraph: *The results presented here, while specific to Bute Inlet, can be relevant to other fjords in the world. Firstly, we argue that any long, deep inlet even with strong freshwater forcing will have a slow return residual circulation, which could contribute to the persistence of subsurface features inside the fjord. The latter could be particularly relevant if there is a potential source/release of contaminants below the surface outflowing layer. Secondly, we note that katabatic wind events are not specific to Bute Inlet, but have also been observed in other fjords of BC (Pickard, 1961), Alaska (Ladd and Cheng, 2016), southeast Greenland (Oltmanns et al., 2014; Spall et al., 2017), and Antarctica (Forsch et al., 2021). The occurrence of these wind events depends on the location, geography, and topography of the fjord (i.e., a fjord must be connected to the continental plateau to experience these wind events). Thus, our findings from Bute Inlet (regarding the slow circulation that allows for the persistence of subsurface water mass) could be representative of deep, long fjords in the mid-latitudes that experience katabatic winds or deep-mixing events in winter.*

Fig 1

Lon/lat are exchanged

Consider using color to show the resolution and move panel (a) to the appendix

Why do the two maps appear different – is the aspect ratio not the same?

Consider including a length scale in (b) to help the reader.

What about showing bathymetry rather than resolution?

We have fixed lat/lon labels and coloured the model grid according to the resolution – thank you R1 for the suggestion. We increased the size of the figure and ensured that the aspect ratio is the same in both maps. We decided to keep panel (a) in this figure, given that it now shows model resolution and an improved inset that better indicates the location of the study area (Reviewer 2's suggestion).

[Figure]

Fig R1.9. New version of Figure 1.

Figure 1 was updated in the manuscript

Fig 2.

See comment above about the usefulness of these metrics – move to appendix

As discussed in our previous response, we believe that the model evaluation belongs to the main text, while adding information of the evaluation in the upper 100 m

We left figure 2 but amended Table 1 to include metrics for the top 100 m of the water column

Fig 3.

Plot only three profiles – and let us know where each one is from. Especially for modelled temperature they are different. Are these differences there initially, or are they "produced" within the model.

As discussed above (response to L158), we agree with R1 that the modelled T profiles are not perfect matches to the observations and we are committed to not overstate in the manuscript the performance of the model. Below we show the figure requested by R1; we could replace our current Fig 3 by a figure similar to this one, but we argue that the current Fig 3 is informative as is – we just need to improve the main text to make sure that the performance of the model is not exaggerated.

[Figure]

R1.10: Observations (black) vs Modeled (red) profiles of T and S in three locations (see inset). Top panels show June 12 and bottom panels, June 26.

Consider "cutting" the profiles, so that you show the upper layer with a different x-axis than the deeper waters. Using the large scale needed for the upper layer, means that changes in the lower layer are not shown. One alternative could be to include a row showing also initial conditions in the same way. The observed structure in salinity/density above about 100m but below the surface layer appears to be missing in the model. Is this feature not there initially, or do they disappear during the run.

We appreciate R1's insightful comment. Figure R1.11 below shows the modelled vs observed profiles below 10 m, such that subsurface features are shown better. As R1 observed, the salinity and density structure associated to the Tmin layer are also over-mixed (the features exist in the initial conditions, as seen in Fig R1.6). Interestingly, we note that the observed T and S features at the Tmin water mass compensate each other in density, such that the observed feature in the density profile is not as pronounced as in either T or S. Thus, the model is able to better represent the density gradients than either T or S. Furthermore, the model properly represents the pycnocline (even if a bit weaker at the depths of Tmin), which is important for the density-driven circulation in the fjord.

We will discuss the model's limitations to properly represent all features related to the Tmin (temperature, salinity and density) and plan to either "cut" the profiles to highlight the differences at depth or to add Fig R1.11 to the manuscript.

[Figure]

Fig R1.11: equivalent to Fig 3 but starting from -10 m; thus, the x-axes are more appropriate to see the differences between model and observations at depth

Figure 3 was updated to the one shown above in the response to L158, which includes both the full depth + the below-10m zoomed plots.

Fig 4

What happens at about 70 km – and why is this not commented in the ms?   How do you explain the velocities below sill depth?  Consider helping your readers see the four layers.

As shown in Fig R1.4, we will highlight the V~0 by means of a horizontal dashed line in this figure. We have discussed how we will improve the wording regarding the flow below the sill depth (see response to L173 above). Around 70 km we see the effect of tidal mixing over the sill – the region of Discovery Islands (characterized by a complex network of narrow channels and deep fjords) has strong tidal currents (Foreman et al, 2012; Foreman et al, 2015). We will add this information to the manuscript.

The horizontal lines were added to fig 4 and the description was also improved (see response to L170). The text regarding the tidal mixing was added to section 4.1: Outside of the fjord where the outer sill is found (around 70 km from the head of the inlet, Fig. 4c) the velocities showed the effect

of tidal mixing over the sill (Fig. 4a) given the strong tidal currents in the Discovery Islands region (Foreman et al., 2012, 2015).

Figure 4 and 5 basically shows the same thing, right? Maybe you only need one of them?

While the two figures show the same information, we argue that both figures are valuable. Fig 4 allows to clearly see the circulation patterns, including the small horizontal variability (the latter is not seen in Fig 5). In contrast, Fig 5 allows to easily compare the strength of the circulation in the different layers, particularly with the new X axis limits, as suggested by R1 in the next comment (see new proposed figure in Fig R1.5).

Fig 5

For clarity, use a velocity scale suitable for the lower layers – and only give the upper layer outflow velocity as numbers?

Thank you for this suggestion, which we have implemented – please see Fig R1.5

Fig 6

a)Use smaller dots. Not sure this figure is necessary?

We have already reduced the size of the dots. We argue that this panel is useful to identify the cold layer as a distinct water mass.

Please see new Fig 6 below with smaller dot sizes.

b-c) I think you need to include panels showing delta ro/delta N2 from the initial conditions for this figure to be meaningful. And would it not be better to (instead or in addition) compare the changes in density/N2 between the start and the end of the run (In the "end of the run" you'd likely have to average over some sensible period, but I think one could use a number less than 29 days? )

We respectfully disagree with R1. These panels focus on how both simulations differ in terms of their mean density and mean stratification. These are crucial points that we will emphasize and clarify in the main text. In particular, the main point is that the circulation is density driven, such that the reduction of the horizontal density gradient (ie, the surface density increases more in the baseline simulation near the head than it increases near the mouth of the fjord) leads to a slowdown of the overall mean circulation. Please note that the 29-day averaging is key to remove the tidal effect from the mean.

New Figure 6:

[Figure]

Figure 6. (a) Temperature-salinity diagram for the baseline (red) and sensitivity (black) simulations in Bute Inlet. Model results shown at the time and location of the observations (12 and 26 June 2019; location in Fig. 4g); for reference, four isopycnals were labelled according to their $\sigma_\theta$ (kg m$^{-3}$). *(b) Profiles of mean density difference between of both simulations ($\Delta\rho$) at 20 and 40 km away from the head of the inlet, shown in the top 100 m of the water column*. Bottom panels show along-inlet transects of the difference in (c) mean density and (d) mean Brunt-Väisälä frequency (N$^2$) between baseline and sensitivity experiments ($\Delta$ = 29-day average baseline minus 29-day average sensitivity; negative values indicate that the baseline is less dense/stratified than the sensitivity simulation).

Table 1

Move to appendix

As discussed in previous responses, we believe that the model evaluation belongs to the main text (potentially adding information on the evaluation of the upper 100m). We can certainly move the table to the appendix, but not its discussion. We are happy to follow the editor's instructions on whether it is better to move the table to the appendix.

We have added the metrics for the upper 100 m into Table 1.

**Reviewer 2**

The MS presents the study of dynamics of intermediate layers in about 730-m deep fjord - Bute Inlet, a mainland fjord in British Columbia, using short-period calculations with a numerical model that was validated by observational data. The used FVCOM finite-element model has variable mesh size from 13 to about 1000 m. The baseline 1-month model run in summer, simulating subsurface temperature minimum due to adopted initial conditions, is complemented by another experiment where the subsurface cold layer was removed from the initial conditions for temperature and salinity. Comparison of the two numerical experiments revealed that layered circulation depends on the initial stratification – usual three-layer flow is replaced to a four-layer one, when cold subsurface layer of Arctic origin is found in the region. The results are this way interesting and worth of publishing.

Reading further, I was not always able to find justifications for the interesting statements.

> Please find below the original response to reviewers in blue, and the corresponding modifications to manuscript in purple.

> We appreciate the thoughtful comments by Reviewer 2 (R2), which will help strengthen our manuscript. We respond to each comment below.

A. "Persistence" is an interesting interpretation that could be discussed, but two one-month model studies do not allow its quantitative evaluation; therefore, this term should be avoided in the title. Two times of this term in the abstract is also not justified. I recommend reformulation of the MS title. Also, the abstract could be rewritten, since about half of it is general introduction not directly connected to the conducted studies.

> We plan to reformulate the focus of the manuscript and improve some explanations, such that the connection between our one-month simulations and the persistence of the subsurface cold water mass is better explained. In particular, we want to make it clearer in the text (and title) that we are not attempting to prove the persistence of the subsurface feature in 2019; the presence of the cold and oxygenated waters from March to October was demonstrated with observations by Jackson et al (2023). Our manuscript does not intend to show that the water mass persisted, but tries to shed light on 'why' it did. Both simulations showed an overall sluggish circulation in this deep inlet, though even slower when the subsurface water mass was present. The slow circulation and long transit times underneath the surface estuarine outflow (calculated to be over 100 days in the baseline experiment at the depth of the temperature minimum; this calculation will be added to the manuscript) are the principal reason why the subsurface feature can remain in place until deep mixing starts in the fall (the latter suggested by Jackson et al 2023 and others).

> In light of all of the above, we will be changing the title of the manuscript to "**Fjord circulation permits persistence of subsurface water mass in a long, deep mid-latitude inlet**" and will adapt the abstract as well (we agree with R2's comment regarding the excess of general introduction and will fix that).

Modifications to the abstract: excess general introduction has been removed from the first half of the abstract. The first couple of sentences now read:

*Fjords are deep nearshore zones that connect watersheds and oceans, typically behaving as an estuary. In some fjords strong katabatic winds in winter (also known as Arctic outflow wind events) can lead to cooling and reoxygenation of subsurface waters, with effects lasting until the following autumn, as observed in 2019 in Bute Inlet, British Columbia, Canada. We used high-resolution, three-dimensional ocean model summer simulations…*

B. Instead of sufficiently long maximally realistic simulation study with variable forcing and boundary conditions, that could describe formation, evolution and decay of intermittent cold subsurface layer, the authors have adopted a simplified approach where open boundary conditions were kept unchanged for a one-month run with modified initial conditions. The authors should carefully justify: (a) Why one-month simulation is appropriate for a process of seasonal duration // from February 2019 to fall, L45-46 //. (b) Why the "sensibility" experiment with altered initial conditions but unchanged forcing is physically feasible. Perhaps it is useful to make an alternative full simulation for the period of missing cold sub-surface layer, when three-layer flow is evident.

(a) As mentioned in the previous response, we are not aiming to simulate the whole process of formation/permanence/destruction of the subsurface water mass. While we understand the reviewer's desire to see a February to October simulation, we note two things. First, we cannot start the simulations earlier, because the high resolution (1km) atmospheric model that provides the surface boundary forcing starts on May 24 2019 (hence, the start date of our model simulations). Second, the deep (~700m), narrow (~2km) inlet is a challenging environment to model in a sigma-coordinate framework; if we had enough observations to do data assimilation, we could likely fix some issues that tend to deteriorate longer simulations. Nevertheless, the simulated summer month allows us to explore the main features of Bute Inlet's circulation and we would not expect large changes further into the summer. Furthermore, we would like to highlight that our one-month simulations show the summer mean circulation at a time-scale similar to many observational studies (e.g. Baker and Pond, 1995; Gillibrand et al, JPO, 1995; Stacey and Gratton, JPO, 2001). Therefore, we argue that the information provided by these simulations is valuable.

(b) The sensitivity experiment aims to show the May/June conditions if the winter Arctic outflow event (or any other winter deep-mixing event) had not occurred. The winter event clearly affected the temperature and salinity of our May initial conditions; however, it did not affect the atmospheric and boundary forcing in May/June. In other words, there is no clear mechanism to explain why/how the winter event would have modified the atmospheric conditions all the way into the summer. Furthermore, the Arctic outflow event depends largely in the local topography, such that it affected Bute Inlet but not other nearby inlets (e.g. Toba Inlet; see Jackson et al 2023). Therefore, there is no clear connection between the conditions at our open boundaries (Johnstone Strait and Strait of Georgia) and the winter event. Therefore, we argue that it is justified to keep the same atmospheric and boundary forcing as in the baseline experiment, while only changing the initial conditions. We do note that it is unclear how the winter event may affect the river forcing in May/June, so that keeping the same river forcing is an assumption and source of uncertainty.

All of this information will be better explained/added into the methods section of the manuscript. Lastly, while addressing the comments by both reviewers, we looked into a few available temperature profiles in Bute Inlet (including those presented by McNeill 1974 and Pickard 1961) and realized that all showed some kind of temperature minimum in May/June, even if very small; therefore, we will make clear in the manuscript that our sensitivity experiment does not necessarily represent a year without an Arctic outflow event, but rather a situation where no deep winter mixing occurred.

Modifications of the text, section 2.3 (new/modified text in *italics*):

*Moreover*, HRDPS-1km outputs were available starting in 24 May 2019 *(i.e., limiting any potential earlier start date)*. The total length of the simulation allowed for 5 days of spinup and 29 days for analysis; the latter is an appropriate averaging period to remove tides and calculate residual flows (Foreman et al., 1992). *Longer simulations were not pursued partly because of the diffusive nature of FVCOM, which makes it challenging to reproduce a deep and narrow fjord without data assimilation. Nevertheless, the chosen month properly represented the summer conditions in the inlet, since the freshwater forcing was high from late May to late September, the wind conditions were stable (blowing mostly from the south until September), and the values at the open boundaries were similar throughout the summer (see Appendix B).*

*To represent the idealized summer conditions in the absence of strong deep winter mixing the previous winter (e.g., by an Arctic outflow wind event), a* sensitivity experiment was performed by removing the temperature minimum feature in Bute Inlet from the initial conditions. *This experiment represents an extreme scenario, given that strong katabatic wind events are common in winter in this region (more than 2 events per year on average; Jackson et al., 2023), such that some degree of subsurface cooling is usually present (e.g., MacNeill, 1974; Pickard, 1961). All other initial conditions and forcings (e.g., atmospheric and open boundaries) remained unchanged, given that the winter deep-mixing event would only affect summer conditions in the fjord (e.g., summer open boundary conditions in the Strait of Georgia and Johnstone Strait would not be affected by the outflow winter event in Bute Inlet). It is unclear how the winter event might have affected the summer river discharge; we kept this forcing unchanged to focus on the role of the initial conditions, acknowledging this assumption is a source of uncertainty.*

C. The paper could reproduce and/or elaborate the observational background of Arctic outflow, the main headline of the MS, and its response in the Bute Inlet, in order to support the present modelling study. There are general papers by Jackson et al. (2022) and (2023) referenced, but meteorological and oceanographic anatomy of the modelled period would be nice to be read from this paper.

We appreciate R2's point of view, but we believe that modelling the evolution of the subsurface water mass from formation to destruction is beyond the scope of this manuscript. We hope that the new proposed title and change of focus make the goals of the manuscript clearer. Furthermore, for the reasons detailed before (i.e., lack of atmospheric forcing data), we would be unable to model the formation and initial stages of the subsurface water mass (February to late May 2019).

Minor remarks

The terms "baseline" and "sensitivity" are commonly used in other meanings than here, please consider reformulation.

> We found this perspective very interesting, since we believe that both terms are currently being used within the standards of the discipline (ocean modelling). We are open to change them if the editor or R2 have specific suggestions. For now, we would keep them as is, since we prefer these terms than the more generic "Experiment 1" and "Experiment 2" type of nomenclature.

1 does not reflect the location of study area in wider geographical context, it was not easy to find it e.g., from Google Maps.

> We have improved figure 1 (including some suggestions from R1). It is now larger and highlights more geographical features, like "USA", "CANADA", and "Pacific Ocean".  Please see below

[Figure]

New Figure 1.

L191: "strong winter mixing event" is introduced, but it remains uncovered (see also conclusions L243 and L247).

The strong winter mixing event that led to the subsurface water mass was described by Jackson et al (2023); as mentioned before, the formation of this subsurface feature is not the focus of this paper. Nevertheless, we plan to improve our description of this event and the creation of the subsurface feature (as described by Jackson et al 2023), such that it is clearer to the reader what conclusions are derived from our analysis (and which ones belong to the existing literature).

Modifications to the introduction (around new line 50; new text in *italics*):

Despite the observed warming and deoxygenation at depth, recent observations in Bute Inlet showed that subsurface (i.e., above the sill depth) cold and oxygen-rich waters originated during an Arctic outflow wind event in February 2019 and persisted until the following fall (Jackson et al., 2023). *These authors discussed how the strong, cold February katabatic winds mixed the top ~100 m of the water column in Bute Inlet, leading to cooling and oxygenation down to that depth; they also observed temperature minima and oxygen maxima in monthly vertical profiles from March until October 2019.*

L247: "Our study highlights how a fjord's circulation can be changed for the whole year by an extreme wind mixing event in winter." Where this statement comes from?

We agree with R2 that this statement was not accurate, since this is not something we really showed. We will remove this statement and, overall, make sure that statements in the discussion/conclusions are properly backed up by our analysis.

We have removed this statement.

---

## Author Response (AR2)

We appreciate the positive comments by the editor and one reviewer. We have changed the title following their suggestions as well as replaced the word "persistence" in five other locations in the manuscript to favour more subjective terms (e.g., lingering, persistent). We have also changed the styling of the regression lines in figure 2a and 2b (from solid cyan to dashed black) to make the plot colour-blind safe.

---

## Author Response (AR3)

We are grateful to the editor for the prompt acceptance of our revised submission. The only changes in this final document are the following

1) Added middle initials to the names of two co-authors (Krassovski and Callendar).
2) Changed the colour and style of the cyan regression line in Fig 2 (now it is dashed black) to make the figure colour-blind safe, as promised in the previous submission.
3) Updated the Data Availability section to reflect the Zenodo dataset and added the reference to the dataset in the reference list. Note: the datasets will be also stored within the Canadian Open Government Portal, but that repository does not provide DOIs.